# Taxonomy, Systematics and Evolution of Giant Deer *Megaloceros Giganteus* (Blumenbach, 1799) (Cervidae, Mammalia) from the Pleistocene of Eurasia

**Roman Croitor**

Laboratory of Terrestrial Vertebrates, Institute of Zoology, Ministry of Culture, Education, and Research, Academiei Str. 1, MD-2028 Chisinau, Moldova; roman.croitor@zoology.md

**Abstract:** The article presents a preliminary morphological description of the holotype of *Megaloceros giganteus* (Blumenbach, 1799) that serves for the description of the species. The article proposes a taxonomical and morphological revision of the nominotypical subspecies *M. giganteus giganteus* and morphological comparison with other subspecies of *M. giganteus*. The cluster analysis of diagnostic craniodental and antler characters revealed the systematic position and phylogenetic relationships of *M. giganteus* with other cervid groups. The genus *Praedama* is regarded as a closely related phylogenetic branch that linked to the direct cursorial forerunner of *Megaloceros* that evolved in the middle latitudes of Western Siberia and northern Kazakhstan. The genus *Dama* has a distant relationship with *Megaloceros* and represents an earlier phylogenetic branch that evolved in the Ponto-Mediterranean area. The article discusses the secondary adaptations of *M. giganteus* forms to forest and woodland habitats in Europe and general paleobiogeographic features of the *Megaloceros* lineage.

**Keywords:** holotype; taxonomy; antlers; functional morphology; ecomorphology; evolution; paleo-biogeography





## 1. Introduction

Giant deer *Megaloceros giganteus* (Blumenbach, 1799) is one of the iconic fossil species best known to the general public with the longest research history covering more than three centuries. The first scientific report on giant deer was published by Molyneux in 1697 [1] and since then a large number of studies has been dedicated to this remarkable species. Giant deer males grow astonishing large antlers, which are absolutely and relatively largest among fossil and modern deer and represent the most attention-grabbing and intriguing evolutionary specialization of the giant deer. The extremely large antlers of *M. giganteus* aroused a long-lasting debate on the origin of such an "inadaptive" feature and its contribution to the species' extinction [2]. The giant deer was cited by supporters of the now-abandoned theory of orthogenesis as an example of excessive and harmful development of antlers that caused the extinction of their bearers (the so-called "antler-extinction hypotheses" [3,4]).

The large antlers of giant deer were regarded as a perigamic structure with allaesthetic significance or specialized organs of visual intraspecific communication during the rutting period that was supported by sexual selection [3,5–7]. Geist [7] proposed a hypothesis that giant deer large antlers may represent an evolutionary "side effect" corresponding to the increased investment of females in large neonates and production of milk rich in solids. The large antlers of *M. giganteus* served as a demonstration of the positive allometric relationship between the body mass and antler size within a population [8]. The giant deer antlers have been a subject of eco-morphological and physiological modelling studies focused upon the evolutionary importance and physiological limitations of the large antler size that could be one of the factors that caused the extinction of *M. giganteus* [4,7,9,10]. The causes, mechanisms, and chronology of extinction of *M. giganteus* were studied in the

context of the Pleistocene megafauna extinction concept [11–15]. Currently, the giant deer is the only deer species whose extinction has been studied in detail.

In most cases, the evolutionary, ecomorphological, and paleobiologic studies of the giant deer are based on the exceptionally rich sample from Ireland [8,9,11,16,17], a circumstance that may be regarded as a weak point, since the results obtained and proposed hypotheses have remained outside the systematical and evolutionary context of *M. giganteus* and the important and major part of the information on giant deer ecomorphology and paleoecological diversity has been overlooked. Despite the very early reports on various forms of "continental" giant deer [18], they remain disregarded in paleoecological and evolutionary studies. Such a "restricted" approach has been caused by the unresolved taxonomic issues and a very broad and confusing understanding of the genus *Megaloceros* during the 20th century that included all Pleistocene continental large-sized deer and many dwarf insular forms [19,20].

*Megaloceros giganteus* is currently divided into several subspecies distinguished mostly by the morphology of their antlers [18,21–26]. The type specimen of *M. giganteus* remained unidentified and, therefore, the diagnosis of the nominotypical subspecies (*M. giganteus giganteus*) has been rather vague and unprecise, causing a stalemate situation for the systematic approach in giant deer studies. Van der Made [27] has pointed out that the numerous publications on giant deer taxonomy do not indicate the original type material of *M. giganteus* and his attempts to trace the original publication describing the type material and its geological age remained unsuccessful. The Irish Late Devensian sample is often arbitrarily regarded as a "normal" or "standard" type of giant deer due to a large number of available specimens [8,17,25,26]. However, the exact provenience of many Irish fossils is unknown and the whole sample of giant deer from Ireland is not homogenous: there are two morphological types of giant deer unrelated with sexual dimorphism and distinguished by mandibular proportions [28] and metapodial length [27,29]. The presence of two closely related, but distinct forms of giant deer in the Devensian of Ireland was recently confirmed by genetic studies [30]. Thus, the identification of the type specimen of giant deer *M. giganteus* becomes very important.

Cuvier [31] has already proposed to designate the well-preserved antlered skull from the "Royal Cabinet" (Musée Nationale d'histoire naturelle, Paris) as a type specimen and provided a detailed description and figures of this skull. In the absence of the original material, this specimen could be regarded as a neotype. However, some characteristics provided by Blumenbach [32] and preceding old reports on giant deer findings permit tracing the specimen that served for the original description of *M. giganteus*. The present study proposes the identification and the preliminary description of the holotype of *M. giganteus* that allowed a precise description of the diagnostic characters of giant deer subspecies. The study also proposes an attempt to reconsider the evolution, origin, ecomorphological specializations of giant deer subspecies, as well as the paleobiogeography of *Megaloceros* and related genera

## 2. Materials and Methods

The study is focused upon the intraspecific morphological diversity of giant deer antlers that provide important diagnostic characters applied in the taxonomy and systematics of this species [18,21,24–26,33]. The study applies the system of antler measurements traditionally used in scientific publications [8,16,21,22,31]. The antler divergence and shape of the antler crown is described by the index of divergence calculated as a ratio between the antler length (the mean length of both antlers, if available) and the antler span. This index provides a general estimation of the presumed adaptation of giant deer forms to a wooded or open landscape and allows the diversity of antler crown shape to be estimated in various forms and samples of *M. giganteus*. Estimated body masses of fossil cervids are based on dental variables according to the method proposed by Janis [34].

A multivariate cluster analysis of the diagnostic cranial, dental, and antler characters is applied to find support for the systematic position of the giant deer and related forms. The hierarchical clustering paired group algorithm UPGMA was computed using the Jaccard

Similarity Index for presence–absence data (PAST-4.03 application [35]). The cophenetic correlation coefficient is computed to estimate how faithfully a dendrogram preserves the pairwise distances among the original, unmodelled data points [36]. The craniodental and the antler characteristics considered in the multivariate analysis are adapted with some modifications and additions from Croitor and Robinson [37] and Croitor et al. [38]. The classification of advanced and primitive conditions for the characters used in the cluster analysis is adapted from Vislobokova [23]. The following morphological characters were considered in the multivariate analysis:

(1) Shape of basioccipital: 0, wedge-shaped; 1, bell-shaped (broadened at the pharyngeal tuberosities);

(2) Braincase length: 0, short (the braincase breadth measured behind pedicles exceeds the distance between bregma and inion); 1, long (the distance between bregma and inion exceeds the braincase breadth measured behind pedicles);

(3) Braincase flexion: 0, unflexed (primitive condition); 1, flexed (advanced condition);

(4) Pedicle length: 0, long (pedicle height exceeds the pedicle diameter; primitive condition); 1, short (pedicle height is smaller than, or equal to, the pedicle diameter; advanced condition);

(5) Pedicle orientation: 0, sloped caudally (primitive condition); 1, set vertically (advanced condition);

(6) Pedicle divergence: 0, little divergent, almost parallel (primitive condition); 1, clearly divergent (advanced condition);

(7) Naso-premaxillary contact: 0, long (primitive condition); 1, short (advanced condition);

(8) Length of the opbito-frontal portion of the skull: 0, short (anterior edge of the orbit is situated above $M^2$ or the anterior part of $M^3$, primitive condition); 1, long (anterior edge of the orbit is situated behind $M^3$, advanced condition);

(9) Length of nasal bones: 0, short (the posterior edge of nasals does not reach the level of the anterior edges of the orbits; primitive condition); 1, long (the nasal bones extend caudally behind the level of the anterior edges of the orbits; advanced condition);

(10) Upper canines: 0, absent (advanced condition); 1, present (primitive condition);

(11) Cingulum in upper molars: 0, absent; 1, present (specialized dental morphology);

(12) Protoconal fold on upper molars: 0, absent; 1, present (specialized dental morphology);

(13) Molarization of lower fourth premolar ($P_4$): 0, unmolarized (primitive condition); 1, molarized (advanced condition);

(14) Antler surface: 0, smooth; 1, pearled (a specialized feature of *Cervus elaphus* and related species and genera);

(15) Position of first antler ramification: 0, low (the ramification height is more or less equal to the antler base diameter); 1, high (significantly exceeds the antler base diameter);

(16) Accessory prong of the first tine: 0, absent (initial morphological condition); 1, present (specialized antler morphology);

(17) Reduction of the accessory prong of the first tine: 0, not reduced or not applicable; 1, reduced (specialized antler morphology);

(18) Reduction of basal tine: 0, basal tine is not reduced (initial morphological condition); 1, basal tine is reduced (specialized antler morphology);

(19) Bifurcation of basal tine: 0, not bifurcated (initial morphological condition); 1, bifurcated (specialized antler morphology);

(20) Additional basal tines: 0, absent (initial morphological condition); 1, present (specialized antler morphology);

(21) Cross-section above basal ramification: 0, circular (initial morphological condition); 1, present (specialized antler morphology);

(22) Shape of basal tine: 0, cylinder-shaped (initial morphological condition); 1, flattened (specialized antler morphology);

(23) Middle tine (trez tine, its homologies and analogies): 0, absent (initial morphological condition); 1, present (specialized antler morphology);

(24)　Posterior tine: 0, absent (initial morphological condition); 1, present (specialized antler morphology);

(25)　Terminal crown tines: 0, absent, 1, present; this type of crown development is characteristic of *Cervus elaphus* and *Megaloceros giganteus*;

(26)　Posterior insertion of crown tines on the beam: 0, absent, 1, present; this crown type is characteristic of *Dama dama* and *Rucervus duvaucelii* that in its simplest variant represents several tines inserted on the posterior side of the beam;

(27)　Anterior insertion of crown tines on the beam: 0, absent, 1, present; among the species included in the study, such position of crown tines is characteristic of *Megaloceros* and *Praedama*;

(28)　Beam curved in the area of posterior tine: 0, no, not applicable; 1, yes (specialized antler morphology);

(29)　Cranial and mandibular pachyostosis: 0, absent; 1, present (specialized physiological feature);

(30)　Comb-like pattern of antler crown: 0, not applicable; 1, clearly expressed.

The studied material includes the giant deer specimens exposed in Barmeath Castle (County Louth, Ireland), the giant deer sample from Ireland stored in the Natural History Museum of London (NHML), the giant deer specimens from the exposition of the National Museum of Natural History, Paris (MNHP), the giant deer sample from Rhine Valley curated at the State Museum of Natural History, Stuttgart (MNHS); and the giant deer specimens and the mounted skeleton curated at Grigore Antipa National Museum of Natural History, Bucharest (GANM); the University of Bucharest (UB), and the Institute of Zoology, Moldova (IZM).

## 3. Systematic Paleontology

### 3.1. Taxonomy

#### 3.1.1. Systematical Context

Family Cervidae Goldfuss, 1820
Subfamily Cervinae Goldfuss, 1820
Genus *Megaloceros* Brookes, 1828
*Megaloceros giganteus* (Blumenbach, 1799)

#### 3.1.2. Synonymy

1799 *Alce gigantea* [sp. nov.]—Blumenbach [32] (p. 197).
1820 *Cervus hibernus* [sp. nov.]—Desmarest [39] (p. 446).
1825 *Cervus megaceros* [sp. nov.]—Hart [40] (p. 23, pls. I-III).
1827 *Cervus* (*Dama*) *giganteus* (Blumenbach, 1799)—Cuvier [31] (p. 306).
1828 *Megalocerus antiquorum* [sp. nov.]: Brookes [41] (p. 61).
1830 *Cervus euryceros* (Aldrovandi, 1621)—Hibbert [42] (p. 301, pl. III, Figure 9).
1834 *Cervus euryceros irlandicus* [ssp. nov.]—Fischer [43] (p. 160).
1838 *Cervus megalocerus* (Hart, 1825)—Fischer [44] (p. 534).
1844 *Cervus* (*Megaceros*) *hibernicus* [sp. nov.]—Owen [45] (p. 237).
1846 *Megaceros hibernicus* (Owen, 1844)—Owen [46] (p. 444, Figure 182).
1892 *Cervus* (*Euryceros*) *hiberniae* (Owen, 1844)—Pohlig [18] (p. 217, Figures 1 and 2).
1929 *Cervus giganteus* (Blumenbach)—Reynolds [16] (p. 1).
1935 *Megaceros euryceros latifrons* ssp. nov.—Raven [47] (p. 178, Figures 1–13).
1953 *Megaceros giganteus* (Blumenbach, 1799)—Azzaroli [19] (p. 48).
1962 *Megaloceros giganteus* (Blumenbach, 1803)—Godina et al. [48] (p. 374, Figure 496).
1987 *Megaloceros giganteus* (Blumenbach, 1799)—Lister [49] (p. 255).
1999 *Megaloceros giganteus* (Blumenbach)—van der Made [50] (p. 398).
2006 *Megaloceros giganteus irlandicus* (Fischer, 1834)—van der Made [27] (p. 125, Figure 10).

### 3.1.3. Identification of Holotype

The original Blumenbach's [32] (p. 697) description of *Alce gigantea* provides quite scanty information about the species, specifying only that this particular kind of "fossil elk" comes from Ireland and is characterized by immense body size. According to Blumenbach [32], the skull of giant deer is almost an ell long and the distance between summits of antlers may attain 14 feet (approximately 4.3 m). The original species description does not provide any reference. Blumenbach [32] specifies in a quite imprecise way the type locality (Ireland) and provides only two measurements, the skull length and the antler span. Nonetheless, the available information is sufficient and allows the identification of specimen with the quoted characteristics.

The publication of Blumenbach [32] was preceded by only five reports of giant deer findings from Ireland that are enlisted here in chronological order: (1) the antlered skull from Dardistown near Drogheda [1]; (2) the shed antlers found near Downpatrick, Northern Ireland [51]; (3) the antlered skull found near Dunleer, Ireland [52]; (4) the antlers found in 1783 near Dromore, County Down, Northern Ireland [31]; and (5) the antlered skull found near the Nobber Village, Ireland [53]. The antlered skull from Dunleer environs that Wright [52] studied when he visited Barmeath Castle (Figure 1) represents a particular interest, since its measurements—namely the unusually large distance between the antler summits—fully correspond to the antler span measurement quoted by Blumenbach [32]. Therefore, following Articles 72.4.1.1 and 72.4.5 of the International Code of Zoological Nomenclature, the antlered skull from Dunleer environs is designated here as the holotype of *Megaloceros giganteus* (Blumenbach, 1799). Apparently, Blumenbach [32] was aware of other specimens discovered in Ireland at that time and they potentially may be considered as paratypes. However, we have no any indication of which other specimens were known to Blumenbach [32]. The holotype of *Megaloceros giganteus* (Figure 2) is currently exposed in Barmeath Castle near Dunleer, County Louth (Ireland). The environs of Dunleer where the type specimen of giant deer was discovered [52] should be considered as the type locality of *M. giganteus* (Blumenbach, 1799).

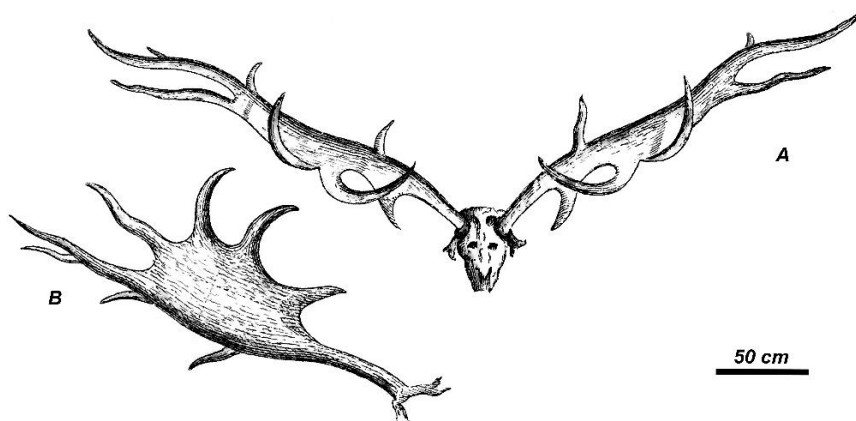

**Figure 1.** Original figure of the holotype of *Megaloceros giganteus* (Blumenbach, 1799) published in "Louthiana" by Wright [52]: (**A**), frontal view; (**B**), lateral view of right antler.

The type specimens of other species names that today are regarded as junior synonyms of *M. giganteus* [25] also may be traced from their original publications. *Cervus hibernus* Desmarest, 1820 [39] is based on several specimens that should be regarded as syntypes: the antlered skull from Dardistown, Ireland [1]; the antlered skull from Cowthrop, England [54]; the shed antlers found near Downpatrick, Northern Ireland [51]; the shed antlers from the environs of Dromore reported by Thomas Percy in 1783 [31]; and the antlered skull found near Nobber, Ireland [53]. *Cervus megaceros* Hart, 1825 is based on the specimen from Rathcannon (Ireland) exposed in the Royal Dublin Society [40]. *Cervus euryceros irlandicus* Fischer, 1834 [43] is based on the specimen described by Hibbert [42] from the Isle of Man that is designated here as the lectotype of *Megaloceros giganteus irlandicus* (Fischer, 1834). The

subspecies name *M. giganteus irlandicus* has been applied by van der Made [27,55,56], but is regarded by Vislobokova [26] as a junior synonym of *M. giganteus giganteus*. Hibbert [57] proposed to use the species name *Cervus euryceros* taken from Aldrovandi's interpretations of antique texts [58]. However, "cervus euryceros" mentioned by Aldrovandi [58] is unavailable for zoological taxonomy [21]. *Cervus* (*Megaceros*) *hibernicus* Owen, 1844 could be taken as based on the specimen described by Molyneux [1], since this is the only specimen that can be undoubtedly identified from Owen's [45] publication.

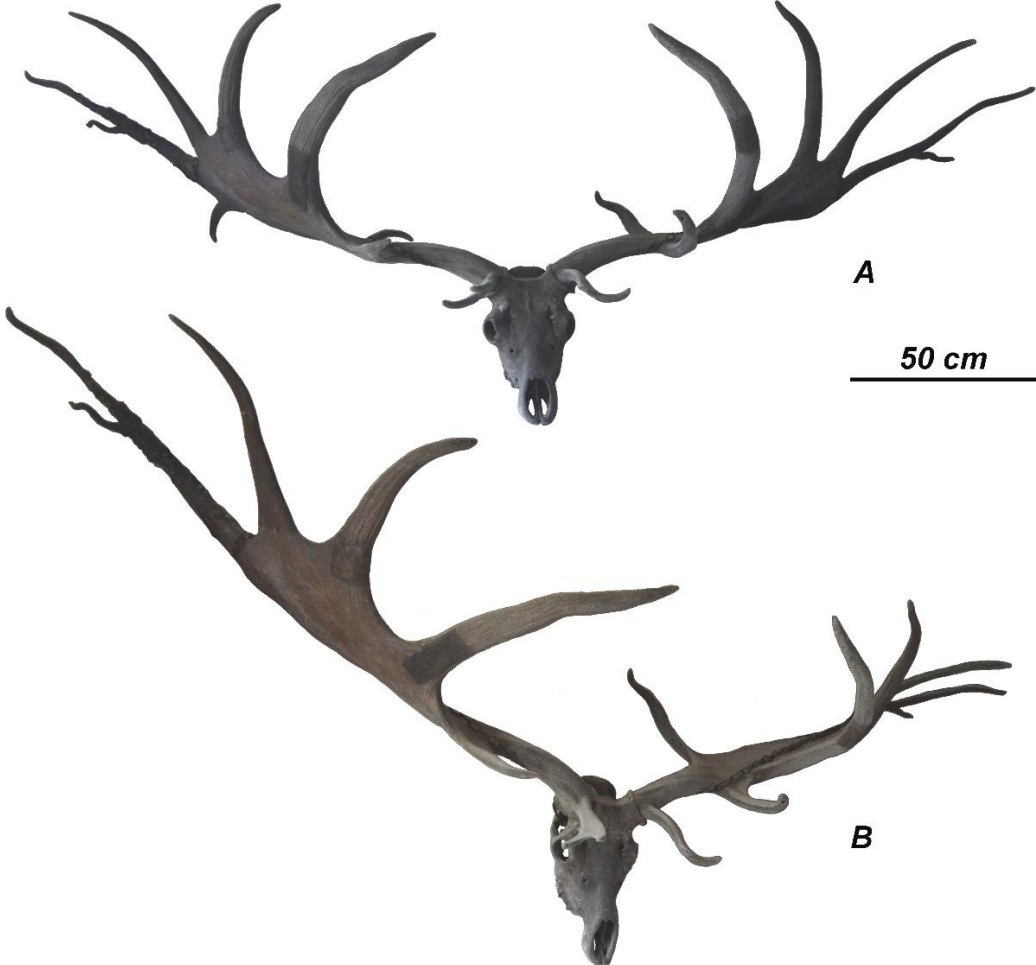

**Figure 2.** The frontal (**A**) and oblique right (**B**) views of the holotype of *Megaloceros giganteus* (Blumenbach, 1799) from Dunleer environs (Ireland).

### 3.2. Description of Holotype

The giant deer from Dunleer environs is characterized by strongly divergent antlers that are slightly bowed in their proximal part and almost horizontal in their middle portion (Figure 2). The distal crown tines deviate upward and form an angle of approximately 20° with the axis of the middle part of the antler. The distal portion of antlers forms a well-developed broad palmation. The antler shape corresponds to the seven-tined bauplan basic for *Megaloceros* (Figure 3A) composed of the following elements: the flattened bifurcated basal tine situated very close to the burr, the moderately large middle tine, the well-developed posterior tine, which is longer and stronger than the middle tine, the two crown tines that are inserted on the anterior border of the palmation; and the three distal tines. The middle distal crown tine in the specimen from Dunleer is bifurcated, thus the total number of antler branches is eight. Posterior crown tines are absent. The basal tine is

S-shaped in the lateral view, with a horizontal orientation of its flattened and bifurcated distal part. The basal tine is roughly as long as the middle tine.

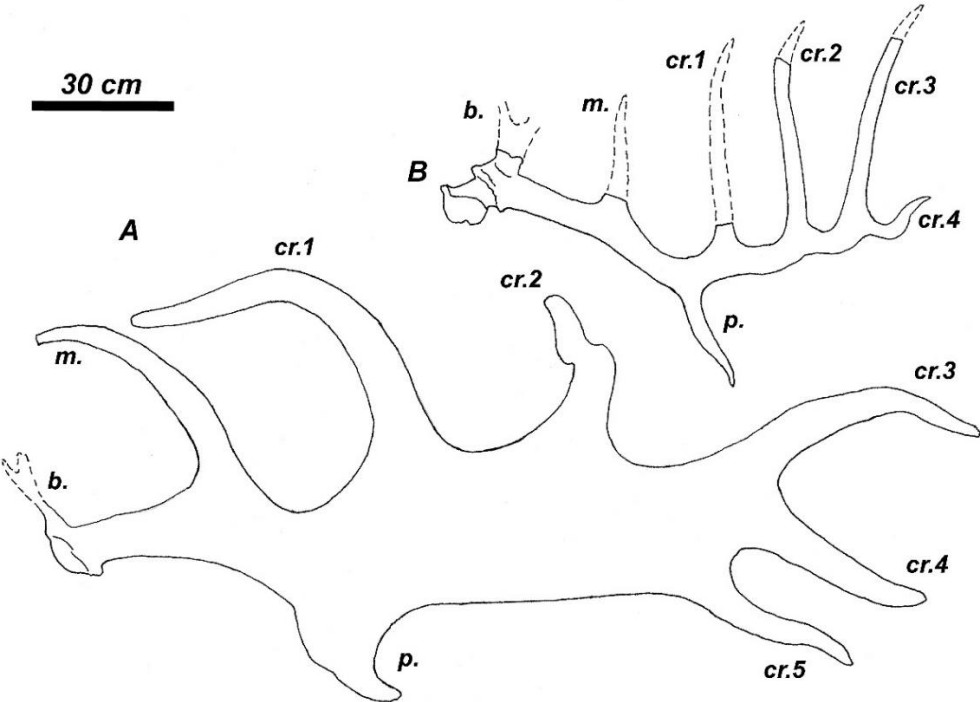

**Figure 3.** Antler bauplan of *Megaloceros* and *Praedama*: (**A**), *Megaloceros giganteus giganteus* from Bucharest environs, Romania (adapted from Apostol [59]); (**B**) *Praedama* sp. from Pinedo, Spain (adapted from Aguado [60]). Abbreviations: b., basal tine; m., middle tine; p., posterior tine; cr., crown tine.

The antler palmation, according to the measurements provided by Wright [52], is the broadest among the specimens and samples involved in the present study: the ratio of palmation breadth to distance between the middle tine and distal edge of palmation attains 65.2%. The palmation of the type specimen is much broader if compared to *M. giganteus* from Kamyshlov (Yekaterinburg, Western Siberia; 27.1% [61]) and Bucharest (26.3% [59], Figure 3A), and broader than in Cuvier's specimen (42% [31]) and the mean value of the sample from Ballybetagh (approximately 45% [8]).

The length of antler palmation in the type specimen from Dunleer (813 mm) just slightly exceeds the mean palmation length in the sample from Ballybetagh Bog (808 mm [8]). However, the holotype of *M. giganteus* is characterized by the longest distance from the antler burr to the middle tine (441 mm) among the considered specimens and samples. The mean distance between the basal tine and the middle tine in the Ballybetagh sample is approximately 370 mm according to Gould [8], or even shorter and in most cases varies between 20 and 30 cm [17]. The difference in the obtained measurements, most probably, resulted from the fact that Gould [8] based his observation on the strongly biased sample toward large antlers. Some antlers from Siberia have similar long distance between burr and middle tine to the Dunleer specimen, as, for instance, the antler TPI-77 from Krasnyi Yar, which has the distance between the burr and the middle tine measuring 410 mm [62].

The ratio between the distance between burr and middle tine and the distance between the middle tine proximal edge to the distalmost point of palmation (the distal crown tines are not included) attains 54.2% in the type specimen from Dunleer. The same ratio is 46.1% in Cuvier's specimen and approximately 46% in the sample from Ballybetagh Bog, 29.1% in the specimen from Yekaterinburg, and 27.8% in the specimen from Bucharest. Thus, the antlers of the giant deer from Dunleer are characterized by the absolutely and relatively longer distance between the burr and the middle tine.

The middle tine of the giant deer from Dunleer coalesces with the palmation. This character is very variable in the Late Pleistocene and Holocene giant deer. The coalescence of the middle tine with the palmation is related to the degree of palmation development. The middle tine is detached from the distal palmation in antlers with poor development of palmation, as in the specimens of giant deer from Kamyshlov [59], Sapozhok (Ryazan, European part of Russia) [63], and Bucharest [59]. The giant deer antlers with strong and broad palmation (the holotype from Dunleer, Cuvier's "neotype", the specimens from Lough Naglack and Knocklong figured by Reynolds [16] show a more or less strong coalescence of the middle tine with palmation. One can assume that the coalescence of the middle tine with palmation resulted from the extension of large well-developed palmation until the middle tine and represent rather a feature reflecting the nutritional conditions during the season the antlers were grown.

The posterior tine in the giant deer from Dunleer is very long and exceeds the length of the middle tine. Gould [8] reported a particularly broad range of variability of the posterior tine in the giant deer from Ireland. The systematical and taxonomical significance of this character is not clear yet. The posterior tine is strong and long in the giant deer from Kamyshlov and Sapozhok that are characterized by a generally weak development of antlers [61,63]. The posterior tine is rather small on the right antler of the antlered skull from the mounted compiled skeleton exposed in the National Museum of Natural History of Paris, but is very long, strong, and bifurcated on the right antler of the same specimen. The posterior tine is rather weak in the specimen studied by Cuvier [31].

The first anterior crown tine in the giant deer from Dunleer is much stronger and longer than the middle tine; it is bowed and its apical part is pointing medially. The second anterior tine is somewhat smaller and arcuate but to a lesser degree.

The exceptionally large antler span of the holotype is partially caused by its extremely long distal crown tines that attain 36.5% of the total antler length and are more or less oriented along the antler main axis. This feature approaches the giant deer from Dunleer to *M. giganteus* from Grigorievka (Pavlodar, northeastern Kazakhstan [64,65]). The distal crown tines of the Bucharest specimen are relatively shorter if compared to the holotype and attain only 22% of the antler length. In the Irish giant deer with compact antlers, the relative distal crown tine length varies from 28% (Cuvier's "neotype") to 17.6% in the specimen from Lough Gur [16] (Figure 8c).

The scatter diagram of the antler span plotted against the relative antler length (Table 1) reveals several types of giant deer according to their antler crown shape (Figure 4). The giant deer from Dunleer with strongly divergent antlers occupies the most extreme position of the diagram. The specimens from Drogheda (Ireland) and Dzhambul (northeastern Kazakhstan) are characterized by similarly strongly divergent antlers, although their absolute antler span is smaller.

**Table 1.** Main measurements (cm) of antlers of *Megaloceros giganteus* and the index of antler divergence (antler length/span of antlers × 100%). Abbreviations: Ssp, subspecies; ID, taxonomic status or collection number; L, antler length; ASP, span of antlers; CFR, beam circumference above the basal tine; IAD, index of antler divergence.

| Ssp | site | ID | ASP | L | CFR | IAD | Sourse |
|---|---|---|---|---|---|---|---|
| *giganteus* | Dunleer | holotype | 426.0 | 243.8 | | 51.7 * | [52] |
| *irlandicus* | Isle of Man | lectotype | 211.0 | 157.5 | 19.7 | 74.6 | [42] |
| *megaceros* | Rathcannon | holotype | 279.4 | 175.3 | | 62.7 | [40] |
| *ruffii* | Cowthrop | | 185.5 | 154.9 | | 83.5 | [54] |
| *hibernicus* | Dardistown | holotype | 330.0 | 157.5 | 20.3 | 47.7 | [1] |
| *ssp.* | Nobber | | 220.0 | 137.2 | | 62.4 | [53] |
| *hibernus* | Dromore | syntype | 300.0 | 222.0 | | 74.0 | [31] |
| *megaceros* | Ireland | NHML, 15282 | 273.7 | 160.4 | 24.8 | 58.6 | [16] |
| *megaceros* | Ireland | NHML, 15602 | 251.5 | 173.4 | 28.0 | 68.9 | [16] |

**Table 1.** *Cont.*

| Ssp | site | ID | ASP | L | CFR | IAD | Sourse |
|---|---|---|---|---|---|---|---|
| *megaceros* | Ireland | NHML, 8085 | 293.4 | 169.9 | 24.1 | 57.9 | [16] |
| *megaceros* | Ireland | NHML, M2323 | 292.1 | 199.4 | 26.7 | 68.3 | [16] |
| *megaceros* | Ireland | Barmeath Castle | 218.0 | 178.5 | | 81.9 | this study |
| *ssp*. | Ireland | NHML, M2324 | 185.4 | 157.5 | 25.4 | 85.0 | [16] |
| *giganteus* | Kamyshlov | Yekaterinburg Museum | 256.0 | 175.0 | | 68.4 | [61] |
| *giganteus* | Sapozhok | PIN-337 | 255.0 | 185.0 | | 72.5 | [63] |
| *ssp*. | Dzhambul | Nr. КΠ 7191 | 350.0 | 170.0 | | 48.6 | [65] |
| *ssp* | Grigorievka | Nr. 582 | 150.0 | 90.0 | | 60.0 | [65] |
| *germaniae* | Bonn | holotype | 160.0 | | 24.0 | | [18] |
| *germaniae* | Worms | | 172.0 | 138.0 | | 80.2 | [18] |
| *antecedens* | Steinheim | MNHS, 15795, holotype | 135.0 | 92.0 | 21.0 | 68.1 | [21] |
| *antecedens* | Steinheim | MNHS, 15925 | 135.0 | 94.0 | 20.0 | 69.6 | [21] |
| *antecedens* | Steinheim | MNHS, 16280 | 210.0 | 115.0 | 25.0 | 54.8 | [21] |

\* Calculated from a photograph.

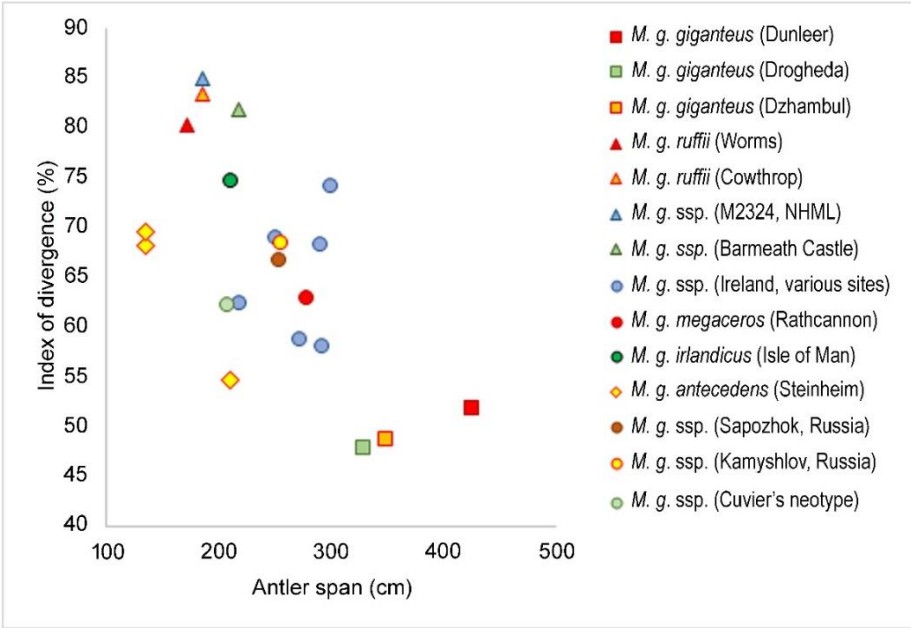

**Figure 4.** Index of antler divergence plotted against span of antlers in various forms and subspecies of *Megaloceros giganteus*.

Cranial sutures of the *M. giganteus* holotype are completely obliterated, however, the position of some of them is still noticeable. Nasal bones are very long and their caudal parts extend far behind the line connecting the anterior edges of the orbits. The profile of frontal bones is concave behind the orbits. Pedicles are robust, moderately divergent (the angle of divergence is 105°) and slightly deflected caudally. Foramina supraorbitale are rather large and situated in well-distinguished supraorbital grooves. Fissurae nasolacrimalis are very small (smaller than foramina supraorbitale), but still clearly visible. Fossae praeorbitalis are practically unnoticeable. The orbitofrontal portion of the skull is rather short: the anterior edge of orbits are situated above the anterior part of M$^3$. Upper canines are absent.

### 3.3. Subspecies

### 3.3.1. *Megaloceros giganteus giganteus* (Blumenbach, 1799)

The nominotypical subspecies is characterized by the most advanced adaptations to open landscape habitats. Its antlers are strongly divergent and directed sideward. The proximal tines (the basal tine and the middle tine) are of moderate size, while the crown tines are very large and long. The distal crown tines are more or less straight and

pointed sideward. The long and strong posterior tine seen in the type specimen from Dunleer, Sapozhok [63], and Kamyshlov [15] may be regarded as a diagnostic feature of *M. giganteus giganteus*. The antler span in large adult specimens normally exceeds 3 m. The taxonomical significance of the extreme degree of reduction of the fossa praeorbitalis in the holotype from Dunleer requires further study.

The association of metacarpal type with antlers based on the material from Ireland is practically impossible since most skeletons (if not all) from museum exhibitions are compiled [8,29,66]. The partial skeleton of the giant deer from Sapozhok (PIN-337 [25,26,63]) is particularly helpful since it provides the association of antlered skull with postcranial skeleton, including a metacarpal bone. Upper molars of *M. giganteus giganteus* are supplemented with a more or less strong cingulum characteristic of late forms of giant deer. The specimen from Sapozhok shows a strong development of cingulum on $M^3$, a fairly well-developed cingulum on $M^2$ and a quite feeble cingulum on $M^1$ [48] (Figure 496a). The giant deer from Sapozhok is characterized by the extremely short upper premolars (the premolar to molar series length ratio is 62.8%) and lower premolars (53.4%), thus achieving the most advanced condition of those characters among the giant deer forms (Figure 5A).

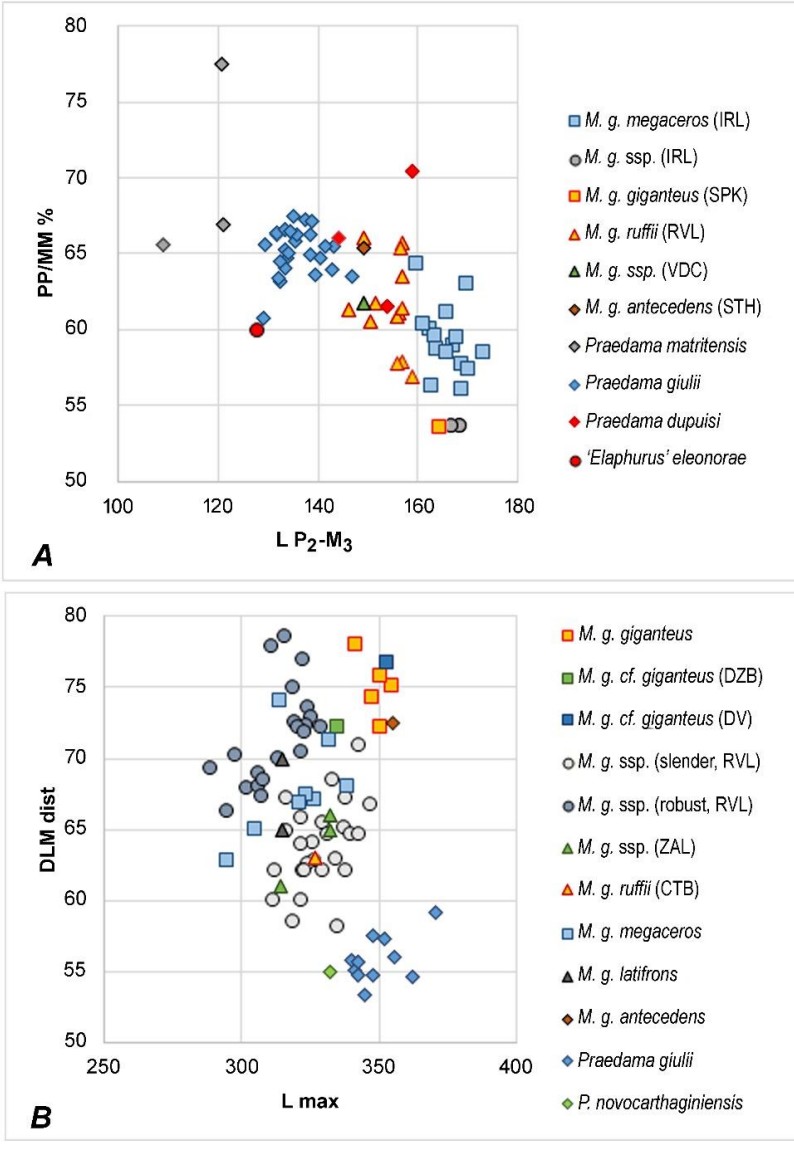

**Figure 5.** Scatter diagrams showing the size and proportions of lower cheek teeth (**A**) and metacarpals (**B**) of various forms and subspecies of *M. giganteus* compared to *Praedama* and "*Elaphurus*" *eleonorae*.

The data on giant deer from Ireland are adapted from Croitor et al. [29]; the data on giant deer from Rhine Valley are adapted from van der Made [27]; the data on giant deer metacarpals from Zhana-Aul are adapted from Kozhamkulova [67]; the data on *Praedama giulii* are adapted from Kahlke [68]; the data on *P. novocarthaginiensis*, *P. matritensis* and "*Elaphurus*" *eleonorae* are adapted from van der Made [56,69] and Vislobokova [70] respectively. Abbreviations: IRL, Ireland; SPK, Sapozhok; RVL, Rhine Valley; VDC, Val di Chiana; STH, Steinheim; DZB, Dzhambul; DV, Duruitoarea Veche; ZAL, Zhana-Aul; CTB, Cottbus.

The specimen from Sapozhok is characterized by the long robust type of metacarpals that are grouped on the scatter diagram together with the long metacarpals from Western Europe (Figure 5B). Therefore, one can assume that the long robust metacarpals from Ireland should be ascribed to *M. giganteus giganteus*. The specimen from the Late Pleistocene of Schlutup near Lübeck (Germany) is also characterized by extremely long and robust metacarpal (L = 351 mm; distal mediolateral diameter, DLM, = 75.8 mm [71]) and therefore is ascribed here to *M. giganteus giganteus*. This form of giant deer was reported by van der Made [27] as the "intermediate type" (*M. giganteus irlandicus*) since its long robust metacarpals have intermediate proportions between the robust and slim types of European giant deer.

The very long and robust metacarpal DV-259 from the multilayered Paleolithic site of Duruitoarea Veche, Moldova (unknown layer), is very interesting, since it may attest to the presence of *M. giganteus giganteus* at this site. Its measurements (L = 353 mm; DLM proximal = 73.5 mm, DLM distal = 76.7 mm) are very close to those of the specimen from Sapozhok and the long-limbed Irish form (Figure 5B). Other remains of giant deer from Duruitoarea Veche are quite old (37,050 ± 450 years BP [15]) and show a morphological affinity with *M. giganteus ruffii* [72].

The degree of palmation development is variable and, as a tissue of low priority, greatly depends on the available forage resources. The palmation attained its maximum degree of development in the holotype specimen, however, it is quite weak and narrow in the giant deer from Sapozhok [63] and possibly reflects stressing environmental conditions. It is necessary to note that the Holocene giant deer from Eastern Europe (Sapozhok) and the South Urals area (Kamyshlov) are characterized by more compact antler crown than the type specimen (Figure 4).

The geography of *M. giganteus giganteus* findings is rather vast. The nominotypical subspecies of giant deer is recorded in Dunleer, Drogheda, Dunshaughlin (Ireland), Schlutup (Germany), Duruitoarea Veche (Moldova), Sapozhok, Kamyshlov (Russia). The antlers from the environments of Bucharest (southern Romania) [59] should also be ascribed to this subspecies. Possibly, some remains of the giant deer from Serbia also belong to *M. giganteus giganteus* [73]. Among the specimens figured by Reynolds [16], only the antlered skulls from the Figure 7a (unknown locality), Figure 9b (Lough Gur), and Figure 11b (Lough Gur) may be ascribed to *M. giganteus giganteus* with a certain degree of confidence. The young adult stag from Lough Beg (Ireland) with strongly divergent but rather short antlers (the antler span attains 266.7 cm [74]) most probably also is *M. giganteus giganteus*.

### 3.3.2. *Megaloceros giganteus megaceros* (Hart, 1825)

I propose to reintroduce Hart's [40] species name to designate the second Late Devensian form of giant deer from Ireland and Western Europe characterized by less divergent antlers and shorter metacarpals. The type specimen from Rathcannon (Ireland) is characterized by a comparatively compact antler crown due to the upward curvature of the distal portion of palmation and rather short distal crown tines that are not oriented along the antler main axis [16,40]. The antler divergence of *M. giganteus megaceros* shows an intermediary condition between *M. giganteus ruffii* and *M. giganteus giganteus* (Figure 4). The antler span rarely exceeds 3 m. The mean antler length attains 117 cm [8]. Unlike *M. giganteus ruffii*, the anterior crown tines are not displaced distally and the palmation is not folded backwards. Most Irish giant deer specimens figured and measured by Reynolds [16] are characterized by the similar to Rathcannon specimen antler morphol-

ogy and the intermediate degree of antler divergence and therefore may be ascribed to *M. giganteus megaceros*. The second skull with smaller antlers from Barmeath Castle (Figure 6) is characterized by the very low index of antler divergence similar to that of *M. giganteus ruffii* (Figure 4), however, unlike *ruffii*, its anterior crown tines are directed toward the anterior and not displaced distally. Therefore, we are dealing in this case with an extreme individual variant of *M. giganteus megaceros*.

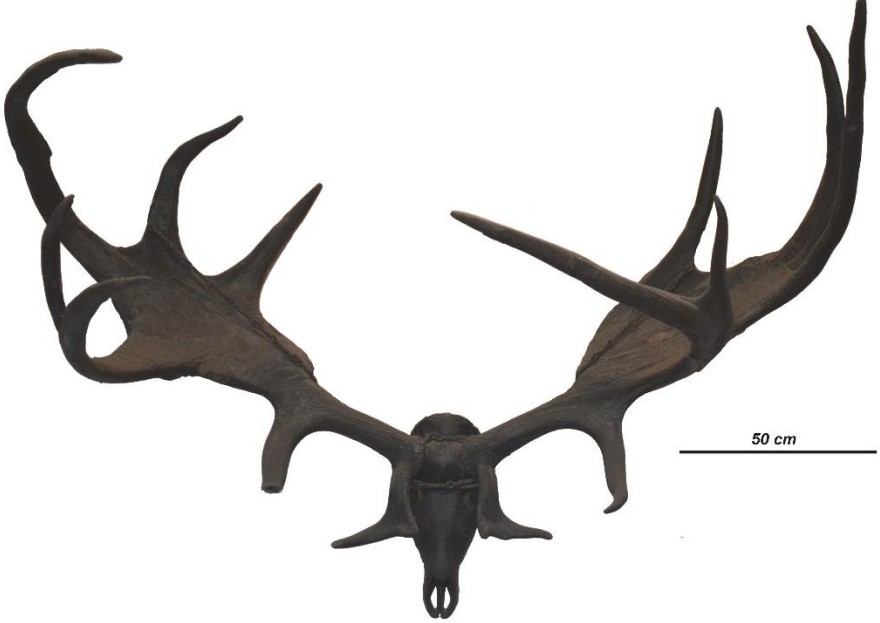

**Figure 6.** The antlered skull of *Megaloceros giganteus megaceros* (Hart, 1825) from Barmeath Castle (Ireland).

The "classical" sample from Ballybetagh Bog probably represents the last population of *M. giganteus megaceros* that survived. The antlers of the Ballybetagh sample are quite small if compared to specimens from other sites. Barnosky [9] describes the Ballybetagh giant deer as a struggling population under critically unfavourable environmental conditions. The cranial material from Ireland stored in NHML shows a strong development of cingulum in upper molars. The upper premolar series of the Irish sample is relatively short (the premolar to molar ratio range is 66.3–76.1%, M = 71.2%, *n* = 11 [72]). This apparent shortening of premolars is a consequence of the relative enlargement of molar series, as, according to Lister [17], the total upper tooth row length in Irish giant deer is enlarged if compared to older forms. The lower premolar series are relatively short (lim = 53.6–66.1%, M = 59.0%, *n* = 20), but possibly the degree of specialization, in this case, is less extreme than in *M. giganteus giganteus* (Figure 5A).

The rather short and thin metacarpals (Figure 5B) that represent the most common type in the material from Ireland may with certain confidence be ascribed to *M. giganteus megaceros*. Van der Made [27] reported this form of giant deer as the "robust type" (*M. giganteus* ssp.) although this form of giant deer did not possess the most robust limbs among European forms of giant deer. The complete skeleton of giant deer from the Isle of Man, possibly, also belongs to *M. giganteus megaceros*. In this case, *M. giganteus irlandicus* (Fischer, 1834) is a junior synonym of *M. giganteus megaceros* (Hart, 1825).

### 3.3.3. *Megaloceros giganteus ruffii* (Nehring, 1891)

This subspecies is geochronologically older than *M. giganteus giganteus* and *M. giganteus megaceros* but characterized by a more specialized shape of antlers that represents an adaptation to forest and woodland habitats [18]. The shape of the antler crown in *M. giganteus ruffii* is generally streamlined due to the palmations curved and twisted medially with the inwardly bowed crown tines (Figure 7). The subspecies is characterized by the comparatively broad flattened basal tine with occasional trichotomy, the frequent

bifurcation of the middle tine, the short beam segment between basal and middle tine, and the vertical and inwardly folded palmation [18,24,25]. The anterior crown tines are displaced distally and adjoin the distal crown tines series. The antler span in *M. giganteus ruffii* is much reduced if compared to *M. giganteus giganteus* and *M. giganteus megaceros* and does not exceed 2 m. The antlered head from Cowthrop (Yorkshire, England) described by Knowlton [54] is characterized by the antler features typical of *M. giganteus ruffii* and should be ascribed to this subspecies. The antler span of the Cowthrop specimen attained only 185.5 cm, while the anterior crown tines, according to the figure provided by Knowlton (1746), are displaced distally. The index of antler divergence and antler span place the specimen from Cowthrop close to *M. giganteus ruffii* from Worms, Germany (Figure 4). The antler length of *M. giganteus ruffii* varies around 140–155 cm [18,54].

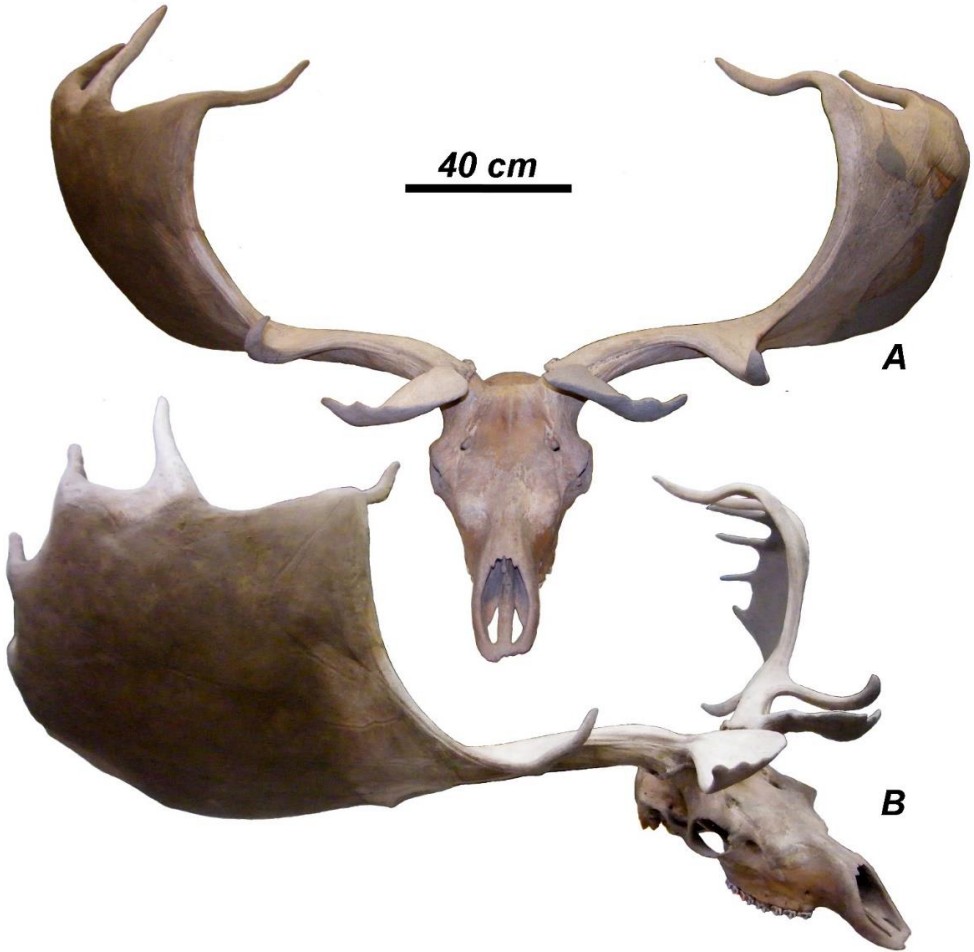

**Figure 7.** The antlered skull of *Megaloceros giganteus ruffii* (Nehring, 1891) from the Rhine Valley, Germany (Nr. 6517.5.9.73.4, MNHS): (**A**), frontal view; (**B**), oblique view. The distal portions of antlers are partially reconstructed.

The cingulum in the upper molars of *M. giganteus ruffii*, as one can see on the large sample from MNHS, is very weak and often only a well-developed entostylid is present on the lingual side of molars. In some specimens (for instance, the skull Nr. 6616.17.7.82.70, MNHS) a weak development of cingulum is observed on $M^3$ and even on upper premolars. Preorbital fossae in males are deep and well developed. Female skulls stored in MNHS are characterized by more shallow preorbital fossae. The upper and lower premolar series are relatively long: the mean value of upper premolar to molar ratio is 74.8% (lim = 71.3–79.6%, $n = 11$, MNHS); the relative length of lower premolar series is 61.5% (lim = 56.9–66.0%, $n = 13$, MNHS).

The complete metacarpal from Cottbus found together with the holotype [75] is rather slender and fits the size and proportions of van der Made's [27] "slender type" of metacarpals from the Lower Rhine Valley (Figure 5B).

*M. giganteus germaniae* (Pohlig, 1892) from Western Germany and *M. giganteus italiae* (Pohlig, 1892) from Italy and Hungary are regarded as juvenile synonyms of *M. giganteus ruffii* [26]. According to Pohlig [18], antlers of the Italian form are more divergent, but this character probably falls within the individual variation of *M. giganteus ruffii* [25].

*M. giganteus latifrons* (Raven, 1935) from the Netherlands may represent another junior synonym of *M. giganteus ruffii*. According to Raven [47], the cranial proportions of *M. giganteus latifrons* have an intermediate position between Pohlig's [18] *hiberniae* and *germaniae* subspecies. The Dutch material includes several fine skulls, but none of them has complete antlers. However, the preserved proximal part of the antler of the type specimen [47] (Figures 1 and 2) shows the twist of the palmed part of the antler specific for *ruffii*. The upper premolar series is relatively long (M = 75.8%; lim = 71.3–80.5%, *n* = 5) and is practically identical with upper tooth series proportions of *M. giganteus ruffii* from other sites. The cingulum in the upper molars is weak. Measurements of the metacarpal "A.h." reported by Raven [47] correspond to the "slender type" from the Rhine Valley, while the metacarpal "U." belongs to van der Made's [27] "short robust type". According to Raven [47], *M. giganteus latifrons* is distinguished by relatively broader skull proportions, however, the subspecies is based on a small craniological sample. One cannot exclude that at least a part of Raven's material belongs to the distinct giant deer form with "short robust limbs".

The area of distribution of *M. giganteus ruffii* ranges over western and eastern regions of Europe, including the Italian Peninsula and southeast Europe north of the Balkan Mountains [18,75,76]. The remains of giant deer from the Paleolithic deposits of Bishnik Cave (Poland) reported as a primitive form of giant deer [29] belong to *M. giganteus ruffii*. The easternmost finding of a fine antlered skull of this subspecies comes from the site of Pushariovka situated in the lower part of Dnieper valley (Central Ukraine) [77]. The remains of *M. giganteus ruffii* appear in the paleontological record with the beginning of the Middle Pleistocene (126 kyr BP) and disappear in western and central Europe with the beginning of the Last Glacial Maximum (25 ky BP) [17,25].

### 3.3.4. *Megaloceros giganteus padanus* (Vialli, 1939)

This endemic Italian form of giant deer originally described as *Megaceros hibernicus* var. *padana* was found together with *Elephas antiquus* and *Rhinoceros mercki* from the alluvial deposits of the Po River near Polesine Parmense (Vialli, 1939) and probably became extinct shortly before the Last Glaciation. This is the most specialized form of *M. giganteus* from Europe. The overall body size of *M. g. padanus* does not differ from the size of other European giant deer: its zygomatic breadth amounts to 224 mm that is very close to the mean value of *M. giganteus ruffii* from Rhine Valley (M = 224.2 mm, lim = 212.0–241 mm, *n* = 11, MNHS), but slightly smaller than in the giant deer from the British Isles (M = 234.8 mm, lim = 220.0–246.0 mm, *n* = 5, NHML) and the sample described as *M. giganteus latifrons* (M = 256.2 mm, lim = 243.0–265.0 mm, *n* = 5, Raven, 1935). The occipital condyles of *M. giganteus padanus* (110 mm) are just slightly broader than the mean value of the male sample from the British Isles (M = 107.7 mm, lim = 101.5–116.7 mm, *n* = 5; NHML, MNHN) and very close to the mean value of the male sample from the Rhine Valley (M = 109.5 mm, lim = 104.7–114.5 mm, *n* = 8; MNHS). The short straight divergent antlers with broad palmations situated at a short distance from the burr are the most striking feature of the endemic Italian giant deer (Figure 8A). The distance between basal tine and palmation attains only 165–170 mm (Vialli, 1935); the palmated portion is also relatively short, so the total span of antlers hardly exceeded 2 m. The middle tine is small and completely fused with palmation, while the posterior tine is very long, strong, and pointed downwards. The shape of the basal tine is unknown, however, the figures provided by Vialli (1939) show that the proximal parts of basal tines were quite broad. The palmations are set almost

vertically, so their surface plane is oriented frontally; the upper portion of palmation with small middle and anterior crown tines are bent backwards. The backward bending of the upper part of palmation is reminiscent of *M. giganteus ruffii* and implies the local evolution of this giant deer form that produced the specialized giant deer *M. giganteus padanus* with shortened antlers that superficially resembles *Sinomegaceros* from eastern Asia.

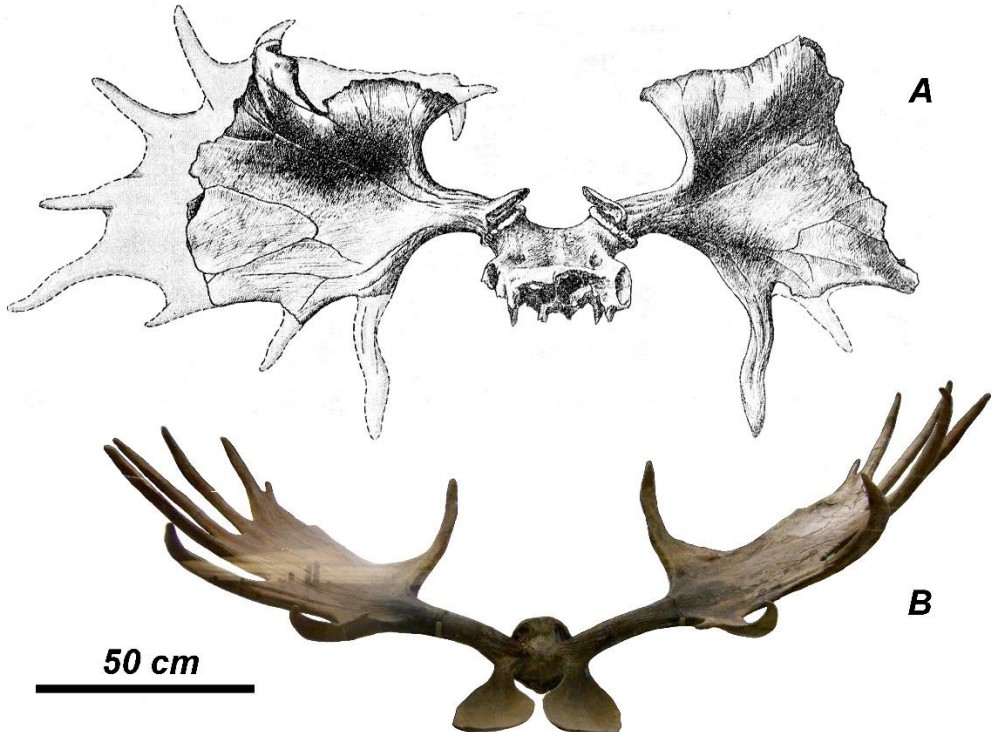

**Figure 8.** (**A**), the type specimen of *Megaloceros giganteus padanus* (Vialli, 1939) discovered near Polesine Parmense, Italy (adapted from Vialli [33]); (**B**), the antlered frontlet of *Megaloceros giganteus antecedens* (Berckhemer, 1941) from Steinheim, Germany (Nr. 16 280, MNHS).

3.3.5. *Megaloceros giganteus antecedens* (Berckhemer, 1941)

The most ancient form of *M. giganteus* in Europe comes from the Middle Pleistocene of Steinheim, Germany (Holsteinian, MIS 11 and 9 [78]. This is also one of the most peculiar subspecies of giant deer that stands apart due to its unusual antler specialization (Berckhemer, 1941). According to Azzaroli [19], the degree of antler specialization allows elevating the taxonomic status of the giant deer from Steinheim to species level. The type specimen Nr. 15795 (MNHS) is characterized by the comparatively insignificant divergence of the quite short antlers with the broad and slightly concave distal palmations and the large plate-like basal tines (Figure 9). The general shape of antlers superficially reminds one of Asian *Sinomegaceros*. However, a closer look allows us to note that the extremely shortened antlers of *M. giganteus antecedens* maintain all typical elements of the generalized giant deer antler bauplan (Figure 10): the middle tine is bifurcated and fused with distal palmation; the two anterior crown tines are longest and strongest among crown tines; the three distal crown tines are variously developed—with the stronger developed anterior tine—but generally are relatively short. The posterior crown tines, as in all other *Megaloceros* forms, are not present. The posterior tine is strong, long and hook-shaped. It is also fused with antler palmation. The specimens of *M. giganteus antecedens* have a marginal position on the scatter diagram (Figure 4), mostly because of their extremely short antlers, while the antler divergence varies from weak to moderate. The difference in antler divergence is regarded as an ontogenetic variation: older individuals of *M. giganteus antecedens* (Figure 8B) are reported to have more divergent antlers [25].

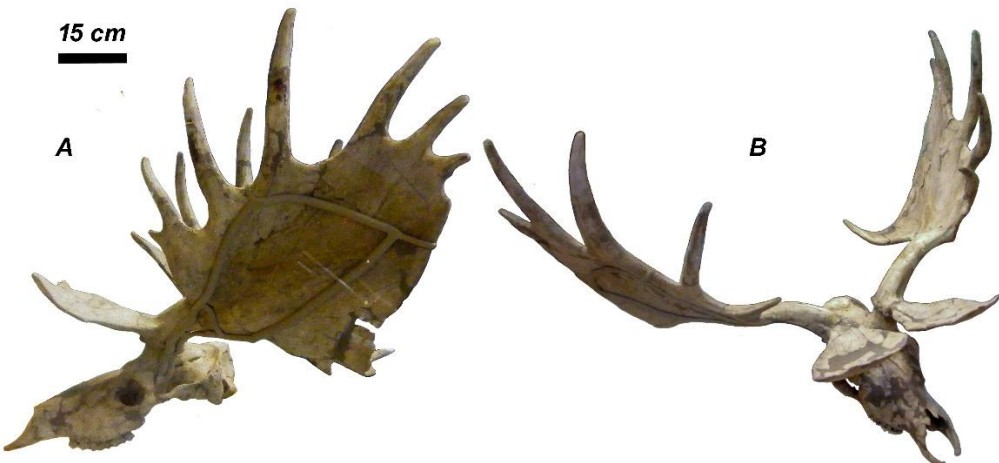

**Figure 9.** The holotype of *Megaloceros giganteus antecedens* (Berckhemer, 1941) from Steinheim (Nr. 15795, MNHS): (**A**), side view; (**B**), oblique view.

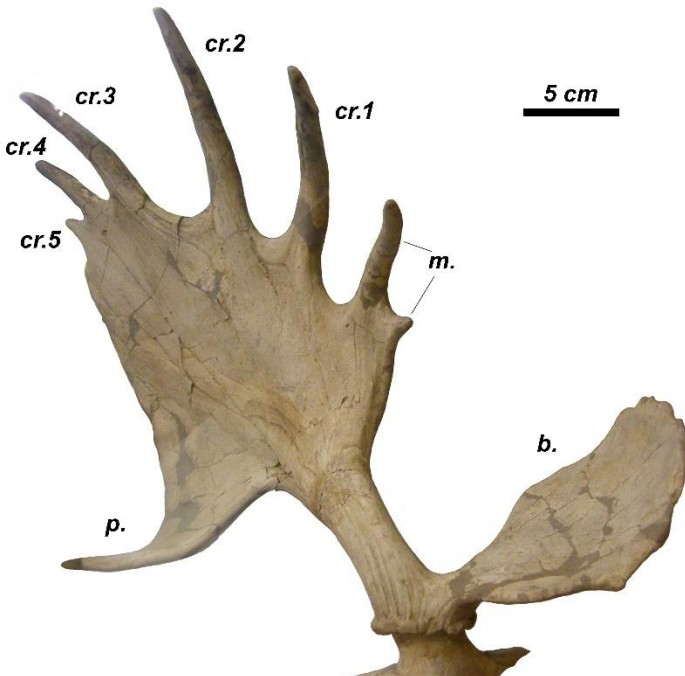

**Figure 10.** Medial view of right antler of the holotype of *Megaloceros giganteus antecedens* (Berckhemer, 1941). Abbreviations as in Figure 3.

The upper cheek teeth are comparatively small (L $P^2$-$M^3$ is 143.7 mm; Nr. 15795, MNHS) with relatively long premolar series (74.0%) as in *M. giganteus ruffii*. The lingual cingula in upper molars are very weak or remain undeveloped. The lingual basal enamel structures are represented in most cases only by flattened leaf-shaped entostyles. The mandible Nr. 16211 (MNHS) from Steinheim is characterized by comparatively small tooth row (L $P_2$-$M_3$ = 149.3 mm; L $P_2$-$P_4$ = 59.0 mm; L $M_1$-$M_3$ = 90.3 mm) and is close to the size of the giant deer mandible from Val di Chiana, Italy [72]. The premolar series of the mandible from Steinheim is one of the longest among the giant deer material involved in the study but still falls within the range of variation of *M. giganteus ruffii* (Figure 5A). The complete metacarpal bone 32806/28 (MNHS) is almost as long as the metacarpals of *M. giganteus giganteus* (Figure 5B). The metacarpal of *M. giganteus antecedens* is quite gracile

if compared to the nominotypical subspecies but still falls within the range of variation of *M. giganteus giganteus*.

## 4. Discussion

### 4.1. Antler Bauplan and Phylogenetic Relationships

Despite the extreme individual and intraspecific diversity [16,18], the antlers of giant deer represent, with some insignificant modifications, the same bauplan that is characterized by the very low position of the basal tine with horizontally flattened and normally bifurcated distal part, the middle tine situated at a certain distance from the basal one, the posterior tine situated on the posterior side of the antler at the level of the middle tine or somewhat more distally, and (most commonly) five crown tines (two anterior and three distal crown tines). There are some individual variants of antlers that occur regularly but do not affect the basic antler bauplan. Sometimes, the additional fourth distal crown tine may be present. The middle and crown tines (and sometimes the posterior tine) may be bifurcated, especially in large and well-developed antlers. The absence of posterior crown tines is one of the most constant features of *Megaloceros* antlers and is considered here as one of the essential features of antler bauplan that evolved during the transition from the three-tined to four-tined evolutionary stages of antlers. The antler bauplan of giant deer is quite conservative and could even represent a certain constrain during the evolutionary transition to wooded habitats in *M. giganteus ruffii* and *M. giganteus antecedens* that show the stronger evolutionary changes of their anterior crown tines. The shape of the distal part of the antler is most frequently affected by evolutionary modifications [17,25]. The proximal part of the antler on the skull directed posteriorly and laterally represents a primitive initial morphological condition of *M. giganteus*, while the more or less lateral orientation of antler beams is characteristic of advanced forms [26].

Several hypotheses on the origin and phylogenetic relationships of *M. giganteus* have been proposed. Owen [46] and Reynolds [16] suggest that giant deer is closely related to modern fallow deer *Dama dama* and, therefore, it is sometimes mentioned as the "giant fallow deer". This oldest hypothesis on the origin of *M. giganteus* found its support in the recent analysis of mitochondrial DNA that indicates the modern fallow deer as the closest, although still rather distant, living relative of *M. giganteus* [79–81]. *M. giganteus* and *D. dama* share such primitive characteristics as the relatively long braincase and the short orbitofrontal portion of the skull, as well as some apomorphies like the long nasal bones that extend behind the imaginary line connecting the anterior edges of the orbits. Both *Megaloceros* and *Dama* have lost their upper canines, a morphological feature shared with *Axis* and *Metacervocerus* [37,82]. Unlike *Dama*, *Megaloceros* maintained such primitive characteristics as the little flexed braincase and a rather oblique position of the pedicles on the skull. The differences in antler bauplan suggest that *Megaloceros* and *Dama* evolved their antler independently and in a different way: in fallow deer, including the earliest known species *D. eurygonos* from the Early Pleistocene of Italy, the crown tines are inserted on the posterior side of the beam; *D. clactoniana* developed crown tines on both anterior and posterior sides of the beam and represents an exceptional apomorphy among *Dama*. The anterior crown tine may occasionally evolve in *D. mesopotamica*, but the development of supplementary occasional prongs in different parts of the antlers represents a characteristic peculiarity of this species [83]. In *M. giganteus*, the crown tines are inserted only on the anterior side of the beam and fringe the distal end of palmation. Therefore, the differences in antler bauplan confirm a relatively distant phylogenetic relationship between *M. giganteus* and modern fallow deer that took place before the evolutionary transition from three-tined to four-tined antlers.

Heintz [84] proposed *Rucervus* (*Arvernoceros*) *ardei* from the Pliocene or Perrier-Etouaires (France) as a probable forerunner of *M. giganteus*, since the *R. ardei* shares with the giant deer such morphological features as the cingulum in upper molars, the accessory prong on the somewhat flattened basal tine, and the development of distal palmation. This viewpoint was accepted by Vislobokova [23]. However, the assumed phylogenetic relationship between

*R. ardei* and *M. giganteus* conflicts with such important features as the general antler bauplan and the relative length of the braincase. Unlike *M. giganteus*, the crown part in *R. ardei* is composed of crown tines inserted on the posterior side of the beam [85]. *R. ardei* is also characterized by such important advanced character as the relatively short braincase, while *M. giganteus* maintains the relatively long primitive proportions of the braincase [37,82,85]. The basal tine with the additional prong in *R. ardei* is not homologous to the bifurcated and flattened basal tine of *M. giganteus*. In giant deer, the range of ontogenetic and individual variation of the basal tine shape passes through the simple flattening without bifurcation (so-called "spoon-shaped" basal tine [74]). This "transitional" variant is never present in *R. ardei*. The craniodental characters and antler bauplan approach *R. ardei* to modern *R. duvaucelii* [85] (Table 2, Figure 11). The results of multivariate analysis obtained in this study put *Panolia eldii*, which is traditionally placed in the genus *Rucervus*, in the cluster together with *Cervus* and related forms in accordance with genetic studies [86].

**Table 2.** Craniodental and antler characteristics used in the hierarchical clustering. Abbreviations: N, number of morphological character (see explications in the research methods description); Mpu, *Metacervocerus punjabiensis*; Mpa, *Metacervocerus pardinensis*; Mr, *Metacervocerus rhenanus*; Ms, *Metacervocerus shansius*; Aa, *Axis axis*; Pe, *Panolia eldii*; Ru, *Rusa unicolor*; Rt, *Rusa timorensis*; Dd, *Dama dama*; De, *Dama eurygonos*; Dc, *Dama clactoniana*; Ce, *Cervus elaphus*; Cc, *Cervus canadensis*; Cn, *Cervus nippon*; Mg, *Megaloceros giganteus*; Ma, *Megaceroides algericus*; Pg, *Praedama giulii*; Ra, *Rucervus ardei*; Rd, *Rucervus duvaucelii*; Rr, *Rucervus radulescui*.

| N | Mpu | Mpa | Mr | Ms | Aa | Pe | Ru | Rt | Dd | De | Dc | Ce | Cc | Cn | Mg | Ma | Pg | Ra | Rd | Rr |
|---|-----|-----|----|----|----|----|----|----|----|----|----|----|----|----|----|----|----|----|----|----|
| 1 | 1 | 1 | 1 | 1 | 1 | 0 | 0 | 0 | 1 | 1 | 1 | 0 | 0 | 0 | 1 | 1 | 0 | 1 | 1 | 1 |
| 2 | 1 | 0 | 0 | 0 | 0 | 1 | 0 | 0 | 1 | 1 | 1 | 0 | 0 | 0 | 0 | 0 | 0 | 0 | 0 | 0 |
| 3 | 0 | 0 | 0 | 0 | 0 | 0 | 0 | 0 | 1 | 1 | 1 | 0 | 0 | 0 | 0 | 0 | 0 | 0 | 0 | 0 |
| 4 | 0 | 1 | 1 | 0 | 0 | 0 | 0 | 0 | 1 | 1 | 1 | 0 | 0 | 0 | 1 | 0 | 1 | 0 | 0 | 0 |
| 5 | 0 | 0 | 0 | 0 | 0 | 0 | 0 | 0 | 1 | 1 | 1 | 0 | 0 | 0 | 1 | 0 | 0 | 0 | 0 | 0 |
| 6 | 0 | 0 | 1 | 0 | 0 | 0 | 0 | 0 | 1 | 1 | 1 | 1 | 1 | 1 | 1 | 1 | 1 | 0 | 1 | 1 |
| 7 | 0 | 0 | 0 | 0 | 1 | 0 | 0 | 1 | 1 | 0 | 0 | 0 | 0 | 0 | 0 | 0 | 0 | 0 | 0 | 0 |
| 8 | 0 | 0 | 0 | 0 | 0 | 0 | 1 | 0 | 0 | 0 | 0 | 1 | 0 | 0 | 0 | 0 | 1 | 0 | 0 | 0 |
| 9 | 0 | 0 | 1 | 0 | 0 | 0 | 0 | 0 | 1 | 0 | 0 | 0 | 0 | 0 | 1 | 1 | 1 | 0 | 0 | 0 |
| 10 | 0 | 0 | 0 | 0 | 0 | 1 | 1 | 1 | 0 | 0 | 0 | 1 | 1 | 1 | 0 | 0 | 0 | 1 | 1 | 1 |
| 11 | 0 | 1 | 0 | 0 | 0 | 0 | 0 | 1 | 0 | 0 | 0 | 0 | 0 | 0 | 1 | 1 | 0 | 1 | 0 | 0 |
| 12 | 1 | 1 | 0 | 1 | 0 | 0 | 0 | 0 | 0 | 0 | 0 | 0 | 0 | 0 | 0 | 0 | 0 | 1 | 0 | 1 |
| 13 | 0 | 0 | 0 | 0 | 0 | 0 | 1 | 1 | 1 | 1 | 1 | 1 | 1 | 1 | 1 | 1 | 0 | 0 | 1 | 0 |
| 14 | 0 | 0 | 0 | 0 | 0 | 1 | 1 | 1 | 0 | 0 | 0 | 1 | 1 | 1 | 0 | 0 | 0 | 0 | 0 | 0 |
| 15 | 0 | 1 | 1 | 1 | 0 | 0 | 0 | 0 | 0 | 0 | 0 | 0 | 0 | 0 | 0 | 0 | 1 | 1 | 0 | 1 |
| 16 | 0 | 0 | 0 | 0 | 0 | 0 | 0 | 0 | 0 | 0 | 0 | 0 | 0 | 0 | 0 | 0 | 0 | 1 | 1 | 1 |
| 17 | 0 | 0 | 0 | 0 | 0 | 0 | 0 | 0 | 0 | 0 | 0 | 0 | 0 | 0 | 0 | 0 | 0 | 0 | 0 | 1 |
| 18 | 0 | 0 | 0 | 0 | 0 | 0 | 0 | 0 | 0 | 0 | 0 | 0 | 0 | 0 | 0 | 1 | 0 | 0 | 0 | 0 |
| 19 | 0 | 0 | 0 | 0 | 0 | 0 | 0 | 0 | 0 | 0 | 0 | 0 | 0 | 0 | 1 | 0 | 1 | 0 | 0 | 0 |
| 20 | 0 | 0 | 0 | 0 | 0 | 1 | 0 | 0 | 0 | 0 | 0 | 1 | 1 | 0 | 0 | 0 | 0 | 0 | 0 | 0 |
| 21 | 0 | 0 | 0 | 0 | 0 | 1 | 0 | 0 | 0 | 0 | 0 | 0 | 0 | 0 | 0 | 0 | 0 | 0 | 0 | 0 |
| 22 | 0 | 0 | 0 | 0 | 0 | 0 | 0 | 0 | 0 | 0 | 0 | 0 | 0 | 0 | 1 | 0 | 1 | 1 | 0 | 0 |
| 23 | 1 | 1 | 1 | 1 | 0 | 0 | 0 | 1 | 1 | 1 | 1 | 1 | 1 | 1 | 1 | 1 | 1 | 0 | 0 | 0 |
| 24 | 0 | 0 | 0 | 0 | 0 | 0 | 0 | 0 | 0 | 0 | 0 | 0 | 0 | 0 | 1 | 1 | 1 | 0 | 0 | 0 |
| 25 | 0 | 0 | 0 | 0 | 0 | 0 | 0 | 0 | 0 | 0 | 0 | 1 | 0 | 0 | 1 | 1 | 0 | 0 | 0 | 0 |
| 26 | 0 | 0 | 0 | 0 | 1 | 1 | 0 | 0 | 1 | 1 | 1 | 0 | 1 | 1 | 0 | 0 | 0 | 1 | 1 | 1 |
| 27 | 0 | 0 | 0 | 0 | 0 | 0 | 0 | 0 | 0 | 0 | 1 | 0 | 0 | 0 | 1 | 1 | 1 | 0 | 0 | 0 |
| 28 | 0 | 0 | 0 | 0 | 0 | 0 | 0 | 0 | 0 | 0 | 0 | 0 | 0 | 0 | 0 | 0 | 1 | 0 | 0 | 0 |
| 29 | 0 | 0 | 0 | 0 | 0 | 0 | 0 | 0 | 0 | 0 | 0 | 0 | 0 | 0 | 1 | 1 | 1 | 0 | 0 | 0 |
| 30 | 0 | 0 | 0 | 0 | 0 | 0 | 0 | 0 | 0 | 0 | 0 | 0 | 1 | 0 | 1 | 0 | 1 | 0 | 0 | 0 |

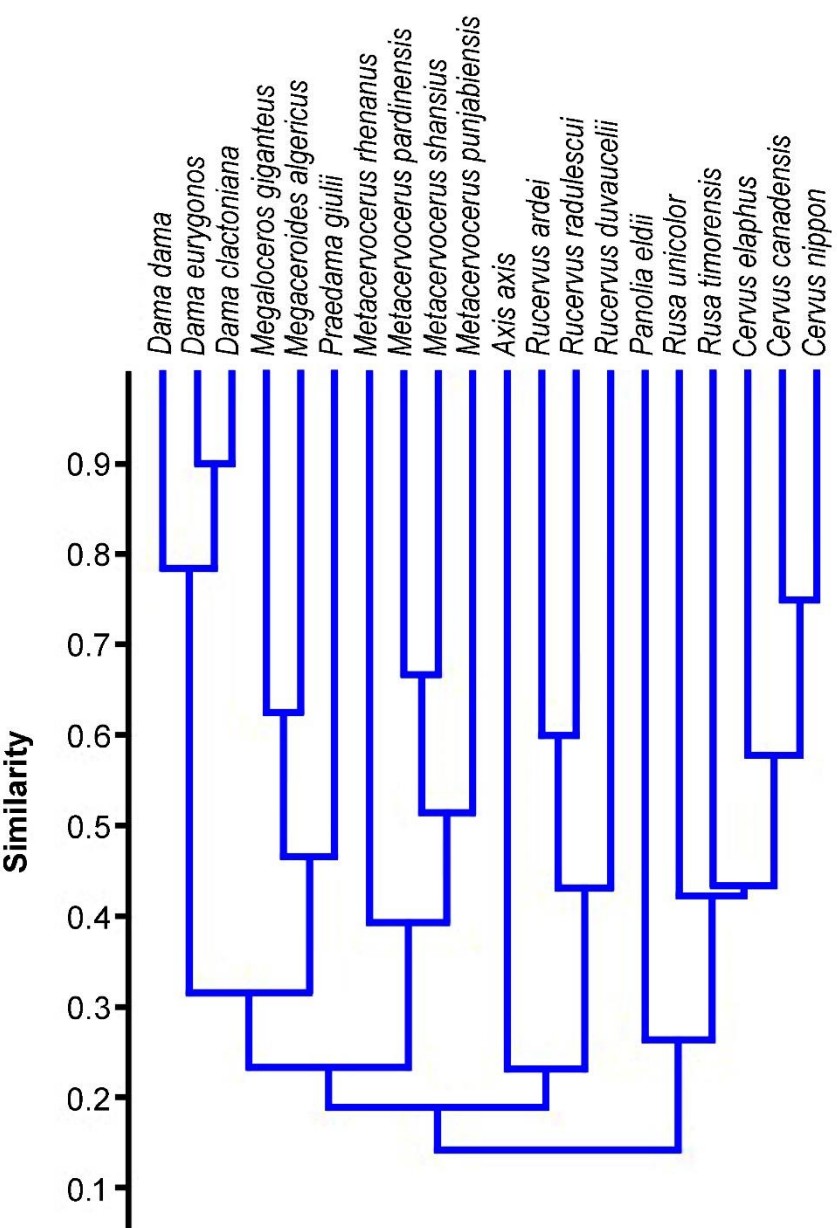

**Figure 11.** Cluster analysis of the diagnostic antler, cranial, and dental characters of *Megaloceros giganteus* and related species, as well as selected modern and extinct representatives of the subfamily Cervinae. The cophenetic correlation coefficient is 0.8342.

Azzaroli [19] assumed the close phylogenetic relationship between *M. giganteus* and *Praedama savini* (Dawkins, 1887) (=*Dolichodoryceros suessenbornensis* Kahlke, 1956) from the Middle Pleistocene of Western Europe. *Praedama savini* is a medium-sized cervid with long thin antlers bearing a flattened basal tine, a middle tine, a posterior tine, and three crown tines as one can see in the well-preserved antlers from Süßenborn [87,88]. The dental morphology is generally unspecialized, with primitive unmolarized $P_4$ and occasional presence of small lingual cingulum in upper molars [89]. The phylogenetic relationship between *Praedama* and *Megaloceros* was accepted by van der Made and Tong [90], however, the cited authors doubted that *P. savini* could be a direct ancestral form of *M. giganteus* because of the divergence in mastication adaptations. Vislobokova [23,26,91] regards *Praedama* as a side phylogenetic branch of the *Arvernoceros—Megaloceros* lineage. Van der Made [92] argued that *Praedama* from Cueva Victoria (Late Early—Middle Pleistocene of Spain) is a forerunner of *M. giganteus antecedens*, which was regarded by the cited author as the

most primitive form of giant deer. Later, van der Made [69] regarded the Iberian lineage of *Praedama* as an evolutionary branch that had a sister phylogenetic relationship with *M. giganteus*. However, the demonstration of the phylogenetic relationship between *Praedama* and *Megaloceros* remained unsatisfactory since it was entirely based on a single character, the shape of basal tine [82]. Well-preserved cranial material that could provide important systematic information at the genus level was missing, therefore any well-founded conclusions on phylogenetic relationships of *Praedama* were impossible. The obtuse angle between the horizontal and ascending mandibular rami in *P. savini* indirectly indicates the specific elongation of the orbitofrontal skull portion that represents an advanced feature [23]. This morphological feature rules out the direct ancestral relationship of European *Praedama* with *M. giganteus* which maintains the primitive short orbitofrontal portion of the skull [73,89]. The presence of the middle and posterior tines in *P. savini* similar to *M. giganteus* is not a unique feature and also is recorded in *Praemegaceros verticornis*. As in the case of *Praemegaceros*, the development of the middle and posterior tines may represent parallelism, since, as was already stated above, the middle and posterior tines appeared during cervid evolution several times in different quite distant lineages [93]. The distal portion of *P. savini* is simplified and looks very different from the pattern seen in *M. giganteus*. The crown part of *P. savini* antlers is often described as one or two subsequent bifurcations that form a rather simple crown [26,90]. According to Vislobokova [26], the dichotomous pattern of antler crown in *P. savini* rather reminds one of the antler bauplan of *Sinomegaceros*.

The complete antlers of *Praedama* from the late Middle Pleistocene of Pinedo, Spain [60], is of particular interest since it is characterized by the bauplan that may be considered as initial for the *Praedama* lineage and shows a striking similarity with the antler bauplan of *M. giganteus* (Figure 3). The crown portion of the antlers from Pinedo has a peculiar comb-like construction with four crown tines. The crown tines 1 and 2 (Figure 3B) are homologous with the anterior crown tines in *M. giganteus*, while the two distalmost crown tines are homologous with distal crown tines in *M. giganteus*. Therefore, the antler crown of *P savini* is significantly shortened and simplified if compared to the bauplan of the antlers from Pinedo [60] and most probably represents a further evolutionary specialization. The antlers from Pinedo offers also a fresh perspective on the large deer from the late Early Pleistocene of Untermassfield (Germany) described by Kahlke [68] as *Eucladoceros giulii*.

The large deer from Untermassfeld (the body mass attained approximately 400 kg) is characterized by a rather primitive dentition with relatively long lower premolars and simple unmolarized $P_4$ (Figure 5A) and extremely long limb bones (Figure 5B). Antler material from Untermassfeld is represented by poor fragments and only one complete juvenile antler with a flattened bifurcated basal tine and a distal bifurcation [68]. Kahlke [68] proposed a hypothesized reconstruction of the antler based on available fragments, including a distal fragment that was interpreted as a comb-like crown of *Eucladoceros*. However, the juvenile antler from Untermassfeld does not recall any ontogenetic stage of development in comb-antlered *Eucladoceros*, while the details of dental morphology and some cranial features suggest an evolutionary divergence between *E. ctenoides* and the deer from Untermassfeld that share only the basic for Cervinae primitive characters [94]. The strong flattened and bifurcated basal tine of "*Eucladoceros*" *giulii* that shows its strong development at the earliest stages of ontogenetic development is rather reminiscent of the antler shape of *Arvernoceros verestchagini* David, 1992 from the Early Pleistocene of southeastern Europe [94]. The detailed figure of complete female skull QW 1992/23910 from Untermassfeld published recently by Breda et al. [95] shows a set of interesting morphological characteristics that reveal the systematic position of "*E.*" *giulii*: the upper canines, unlike *Eucladoceros* and *Rucervus* (*Arvernoceros*), are missing; the basioccipital is rather narrow (not bell-shaped as in *Eucladoceros*), the orbitofrontal portion is rather long (the anterior edge of the orbit is situated above the caudal edge of $M^3$) as it was predicted for *Praedama savini*, the nasal bones are moderately long and attain the imaginary line connecting the anterior edged of the orbits, the braincase is little flexed and its relative

length is equivocal but somewhat longer than in *Eucladoceros*: the distance between bregma and inion (=opisthocranion [23]) in the specimen QW 1992/23910 is roughly equal to the braincase breadth. Unlike *Eucladoceros*, the antlers of "*E.*" *giulii* do not possess the supplementary prong in the area of basal ramification [94]. According to Breda et al. [95], the flattened basal tine of antlers from Untermassfeld recalls *P. savini*. It is necessary to note that the distal "comb-like" fragment of antler IQW1982/18587 from Unretmassfeld [68] is very similar to the crown part of the antlers from Pinedo [60]. One can argue that the large deer forms from Untermassfeld and Pinedo belong to closely related species or the same species. Taking into account this assumption, multivariate cluster analysis places the deer from Undermassfeld close to *Megaloceros giganteus* and *Megaceroides algericus* (Figure 11). Therefore, the large long-limbed deer from Untermassfeld is included here in the genus *Praedama* as *Praedama giulii* (Kahlke, 1997). The branch *Praedama-Megaloceros-Megaceroides* is associated with *Dama* and shows a more distant affinity with *Metacervocerus*, which, in turn, has a sister relationship with modern *Axis* and *Rucervus* (Figure 11). The results of cluster analysis are interesting since they indicate *Metacervocerus* as a genus representing the early evolutionary radiation leading to *Dama*, *Praedama*, and *Megaloceros*. It is necessary to mention that some of the representatives of *Metacervocerus* are characterized by some specific features that we find in *Dama* and *Megaloceros*. This is the case of *M. pardinensis* characterized by the presence of cingulum in upper molars [84] and *M. punjabiensis* that maintains an elongated braincase as in *Dama* and *Megaloceros* [37].

The large-sized deer from Rosieres described by Stehlin [96] as *Cervus* (*Megaceros*) *dupuisi* also should be included in the genus *Praedama*. Stehlin [96] noted the specific robustness of the mandibles of the deer from Rosieres and suggested that this feature approaches *Praedama dupuisi* to *M. giganteus*. The size of the deer from Rosieres is relatively small (L $M_1$-$M_3$ is 82 mm [96]), but falls perfectly within the range of variation of *P. giulii* (L $M_1$-$M_3$: M = 83.4 mm, Lim = 91.0–80.0 mm [68]). Among other specific characters of the deer from Rosieres should be mentioned the primitive unmolarized $P_4$ and the flattened basal tine [96]. Unfortunately, the sample from Rosieres does not contain complete limb bones that could be useful for comparison with *P. giulii*. According to Stehlin [96], *Cervus* cf. *dupuisi* from Süßenborn, is closely related to the deer from Rosieres, but shows more advanced "megaceroid" features in its antlers. Apparently, Stehlin was referring to the cervid that later was described as *Praedama suessenbornensis* by Kahlke [87,88].

Stehlin [96] also mentions the large-sized deer from Saint-Prest as a cervid form closely related to *P. dupuisi*. Azzaroli [19] confirmed that the deer from Saint-Prest is identical to *P. dupuisi*. Guerin et al. [97] reported the deer from Saint-Prest as *Praemegaceros verticornis*, however, this determination is questionable. The pedicles of the specimen SPP-66 from Saint-Prest are less divergent and maintain the initial cylinder shape, unlike the divergent and compressed rostrocaudally pedicles in *P. verticornis*. The morphology of the basal portion of antlers is completely different to the antler shape of *P. verticornis*: the basal tine in the specimen SPP-66 becomes flattened in its distal portion and has a common insertion on the anterior side of the antler base. In *P. verticornis*, the basal tine is reduced and in most cases is completely lost. The most proximal tine of *P. verticornis* is large, bowed, cylinder-shaped, and situated on the medial (or dorsal) side of the beam [19,28,87,88]. This dorsal tine is a homology of the accessory small prong in *Eucladoceros* [28]. The available morphological features, such as the horizontal flattening of the basal tine, the relatively robust mandibles, the simple unmolarized $P_4$, and the relatively long premolar series suggest that the deer from Saint-Prest is very close or even identical with *Praedama dupuisi*. The size of the lower tooth series of the small sample from Saint-Prest (Figure 5A; the measurements used for the diagram are partially estimated) rather corresponds to the size of *M. giganteus antecedens* and *M. giganteus ruffii* and partially overlap with *P. giulii*. The lower premolar series of *P. dupuisi* are relatively long as in *P. giulii* and the measurements. Evidently, *P. dupuisi* and *P. giulii* are closely related and may be synonymous.

The combination of cranial, dental, and antler characters suggest that *Praedama*, as Vislobokova [26] has already proposed, is a side European branch of the phylogenetic

lineage leading to *Megaloceros*. *Praedama* is characterized by primitive dentition combined with some advanced cranial features, such as the elongated orbitofrontal part of the skull, caudally extended nasal bones and moderately short braincase. Taking into consideration the bauplan of the antlers from Pinedo, *Praedama* may be included in the genus *Megaloceros* as a subgenus; however, a detailed revision of the material from Pinedo, Untermassfeld, Süßenborn, and other sites is needed to make a conclusion.

*Praedama* dispersed into western Eurasia by the end of the Early Pleistocene preluding the end-Villafranchian faunal crash. "*Elaphurus*" *eleonorae* Vislobokova, 1988 from the Early Pleistocene of Navrukho (Tajikistan) is mentioned by van der Made and Tong [90] as a cervid that shows a certain morphological affinity with *Praedama* and *Megaloceros*. This is a rather large cervid similar in size to modern European red deer. It is characterized by quite simple antlers with horizontally flattened bifurcated basal tine and a poorly branched distal portion. The middle and posterior tines in "*E.*" *eleonorae* are not present [70]. The protocone and hypocone in the upper second premolar ($P^2$) of "*E.*" *eleonorae* are separated by a deep split [70] as in *Praedama* [56,69] and *Megaloceros* [29]. However, unlike *Praedama* and *Megaloceros*, the hypocone of upper premolars in "*E.*" *eleonorae* do not evolve inner enamel folders. "*E.*" *eleonorae* has already achieved the evolutionary advanced state of short lower premolars (Figure 5A) and, therefore, cannot be regarded as a direct forerunner of *Praedama* and *Megaloceros*. "*Elaphurus*" *eleonorae* with flattened and bifurcated first tine may support the hypothesis on the origin of *Praedama* and *Megaloceros* in the middle latitudes of central and western Asia, however, the missing middle tine in the Tajik deer does not fit this hypothesis. Vislobokova [23] approached "*E.*" *eleonorae* to *Cervus* (*Elaphurus*) *bifurcatus* Teilhard de Chardin et Piveteau, 1930 and included both species in the subgenus *Elaphuroides* Otsuka, 1972.

The combination of primitive dentition with large body size and exceptionally long limbs suggest that *Praedama giulii* was a cursorial browser/mixed feeder adapted to open wooded habitats with tall grass and shrubs. *Praedama* successfully dispersed into the Iberian Peninsula where it was described as *Praedama novocarthageniensis* (=*Megaloceros novocarthageniensis* van der Made, 2015 [56]) and was reported from the Early Pleistocene of Libakos (Greece) as *Megaloceros* aff. *savini* [90]. It is possible that the long-limbed cervid from Madonna della Strada and Selvella (Italy) described as *Arvernoceros* sp. [98] also represents the successful dispersal of early *Praedama* in Europe, however, diagnostic antler remains from the Italian sites are still missing.

*P. novocarthageniensis* from Cueva Victoria is distinguished from *P. giulii* by its somewhat shorter metapodials (Figure 5B) that suggest the loss of cursorial specialization. Further evolution of *Praedama* in Western Europe is marked by the simplification and shortening of the crown part of the antler characteristic for *P. savini* from Süßenborn [87,88]. The latest representative of *Praedama*, *P. matritensis*, survived on the Iberian Peninsula until the late Middle Pleistocene, thus marking the local endemic character of the Pleistocene Iberian faunas [69,89]. *P. martitensis* was a rather small cervid (approximately 150 kg) similar in its body size to modern Iberian red deer. *P. martitensis* maintained such primitive dental characters as the relatively long lower premolar series (Figure 5A) and unmolarized $P_4$ [69]. Among apomorphic characters of *P. matritensis* should be mentioned the strong divergence of pedicles and the mandibular pachyostosis that represents curious parallelism with *M. giganteus*.

### 4.2. Functional Morphology of Antlers and Evolution of M. giganteus in Western Palearctic

The deer of the subfamily Cervinae evolved a great diversity of antlers that traditionally are used as a basis of cervid taxonomy and systematics [19,22,23,82,84,87]. Cervid antlers serve as species-specific organs of communication during the rutting period [99], therefore the broad implication of antler morphology in cervid systematics have a sense. The understanding of functional morphology and eco-morphology of deer antlers have particular methodological importance, as it allows us to estimate the evolutionary significance

and systematic value of antler characteristics. This is especially true for the specialized antlers of *M. giganteus*.

Gould [8] reported a relatively low individual variation of the basal and middle tines in giant deer from Ireland compared to other parts of its antler. This is a very interesting observation that suggests the exposure of the basal and middle tines to a certain stabilizing natural selection due to their vital importance. According to Davitashvili [5], the branched and sophisticatedly complicated antlers in large-bodied advanced cervids represent a specific adaptation that "softens" the sexual selection and diminishes excessive male mortality during intraspecific combat. Such protection of stags against wounding and excessive mortality becomes particularly important in cervids from the middle latitudes where males are facing the unfavourable winter season after the exhausting rutting period [93]. The eye-protection function of the flattened and bifurcated basal tine in giant deer is generally accepted and never has been contested [17]. The basal tine has a locally restricted protection of the face and, in particular, eyes during male combat. In relatively large Palearctic deer, such as *Praemegaceros obscurus* and *Cervus elaphus*, the protecting proximal antler structures become more complicated. In *C. elaphus*, an additional bez tine appeared next to the brow tine, while *P. obscurus* evolved the additional bowed dorsal tine situated above the basal tine [19,28]. It is possible that the bifurcation and flattening of the basal antler in "*Elaphurus*" *eleonorae* represent a similar evolutionary adaptation triggered by the body size increase. This early morphological adaptation has been maintained almost in all lineages related to *Megaloceros giganteus*. *Megaceroides algericus* is the only exception, as this extremely specialized North African endemic deer has completely lost its basal tine [100].

The middle tine represents an additional and broader level of protection of males during the rutting combats: the middle tines lock rival's antlers at a safe distance and increase the safety of both combatting stags in large-sized deer [93,101,102]. Such reinforced two-level protection during combat becomes important in large cervines that attain a certain body mass threshold increasing the risk of lethal wounding [93]. The importance of the middle tine as an evolutionary acquisition in middle-latitude cervids is confirmed by its independent development in several large-sized deer lineages, such as *Cervus*, *Dama*, *Praeelaphus*, the lineages *Praemegaceros obscurus—P. dawkinsi*, *P. pliotarandoides—P. verticornis*, and the lineage represented by *P. solilhacus* [28,93]. The fighting behaviour of stags during the rutting season was very important at the earlier stages of *Megaloceros* evolution [24,26], although the biomechanical study recently carried out by Klinkhamer et al. [100] confirmed a possible constrained fighting behaviour in Irish giant deer too. Thus, the function of protection against rival's antlers applies also to the middle tine of *M. giganteus*. One can assume that the middle tine in *Megaloceros* (as well as *Megaceroides*, *Praedama*, and *Dama*) is a homology of the anterior smaller tine of the second ramification in *Metacervocerus* that, according to the proposed cluster analysis here (Figure 11), represents the three-pointed stage of antler evolution of the *Megaloceros* and related genera lineage.

The posterior tine is one of the most variable elements of antlers in giant deer [8]. The posterior tine also appeared independently in several cervid lineages (*Megaloceros*, *Praemegaceros*, *Sinomegaceros*, *Rangifer*) that are adapted to more or less open environments. The most plausible functional significance of the posterior tine is scratch grooming that diminishes the ectoparasite burden and improves the vigilance and physical state of the rutting stags [93]. The high variability of the posterior tine in the Irish giant deer reported by Gould [8] may be explained by the relaxed evolutionary selection of this feature in the Irish populations or by the mixed character of the studied material.

If compared to the antlers from Pinedo, the antlers of *M. giganteus* evolved a significantly larger crown part, which extended into a palmation (Figure 3). The distal palmation represents a rather late evolutionary acquisition of giant deer and is one of the most evolutionary flexible parts of antlers that provides the most remarkable diagnostic characters of giant deer subspecies [18,21,25,26,33,75]. It is generally accepted that the palmed part of antlers in cervids serves as an organ of visual communication between males and between males and females during the rutting season in the conditions of more or less

open landscape [5,7,8,17,98]. The subspecies of *M. giganteus* are distinguished by such characters of antlers as the relative length of the palmated part of the antler, its bending, twisting, the position of the anterior crown tines, and the size and shape of the antler tines [25,26]. The upright orientation of the distal portions of antlers is regarded as an apomorphy in giant deer [26] and is explained by the adaptation to forest or woodland conditions [7,18]. According to Lister [17], this hypothesis is consistent, since the antlers with an upright position of palmations occur in the interglacial conditions of the Holsteinian and Eemian. Therefore, the antler shape in diverse forms of *M. giganteus* reveals different environmental conditions of rutting and implies important interconnections between environmental conditions, rutting behaviour, visual signals, and ecological conditions of reproduction. The extreme development of sexual dimorphism via the antlers implies large mating congregations, intra-sexual combat, the polygyny, and the combination of female-choice and male-hierarchy with inter- and intra-sexual displays (social status, social competence, health state, and nutritional conditions of a male) [7,8,17]. The outlined natural sexual selection mechanism in giant deer implies that the large size of antlers in *M. giganteus* greatly depends on environmental conditions and cannot be heritable, since in this case antlers lose their communication flexibility and become an inadaptative useless and resource-demanding burden that diminishes the evolutionary competitivity of their bearer with conspecifics with smaller antlers [7].

The general evolutionary trend of *Megaloceros* was shaped by the gradual environmental changes during the Pleistocene toward more open, continental and seasonal environmental conditions in the middle latitudes of central Eurasia that is regarded here as the core area of *Megaloceros* evolution and evolutionary radiation [89]. Nonetheless, the diversity of giant deer and related forms from western Eurasia suggest a more complicated evolutionary scenario.

*M. giganteus antecedens* is the earliest form of giant deer recorded during the Holstein Interglacial in Western Europe. The Steinheim Holsteinian fauna contains such woodland species as *Palaeoloxodon antiquus*, while higher strata of Steinheim (probably late Holsteinian to early Saalian) has yielded the remains of a colder fauna including *Coelodonta antiquitatis* and *Mammuthus primigenius* [103]. During the Middle Pleistocene, northwestern Europe acted as a humid climate refugium for forest species, mostly archaic Palearctic cervids [104]. The early representative of giant deer from Steinheim already possesses such specific for *M. giganteus* apomorphies as the mandibular pachyostosis, the molarized $P_4$, the long nasal bones, and the palmated antlers. *M. giganteus antecedens* still maintains such open-landscape adaptation features as the cursorial long limbs as in *Praedama giulii* and the antler palmations, however, the antlers in *M. giganteus antecedens* are characterized by the secondary shortening of the crown portion and the reduction of antler span as a consequence of adaptation to the forested habitats [17,21,26]. Nonetheless, as already noted, the antlers of *M. giganteus antecedens* maintain all elements typical of an antler of giant deer (Figure 10). Therefore, *M. giganteus antecedens* cannot be regarded as the most primitive form of giant deer since its adaptation to forest and woodland habitats have a secondary and derived character. The extreme antler shortening caused some important morphofunctional changes that finally resulted in the unusual ("*Sinomegaceros*-like") appearance of *antecedens* antlers. The middle tine became very small and adjoined the antler palmation (Figure 10), therefore it could not maintain the initial function of second-level protection during rutting combats through locking rival's antlers at a safe distance. Possibly, the loss of function of protection by middle tine was compensated by the extreme enlargement of the basal tine that was transformed into a large plate-like structure. The antlers of *M. giganteus antecedens* represent functional parallelism with *Sinomegaceros* that, however, is based on a different antler bauplan: unlike *Megaloceros*, the antlers of *Sinomegaceros* initially evolved from the three-pointed antlers with the very low position of the second bifurcation that has been transformed into the distal palmation (Figure 12) [26,38]. Therefore, due to the initial short distance between the antler base and distal palmation and the lack of enough space on the antler beam, *Sinomegaceros* could not evolve the middle tine and arrived

directly to the antler shape with a large palmated basal tine. This circumstance also may explain why *M. giganteus* failed to disperse into eastern Asia. One can assume that the great eco-evolutionary diversity of East Asian Cervinae, including large *Sinomegaceros* that were already adapted to the local woodland conditions, represented an effective biogeographic obstacle for *M. giganteus*.

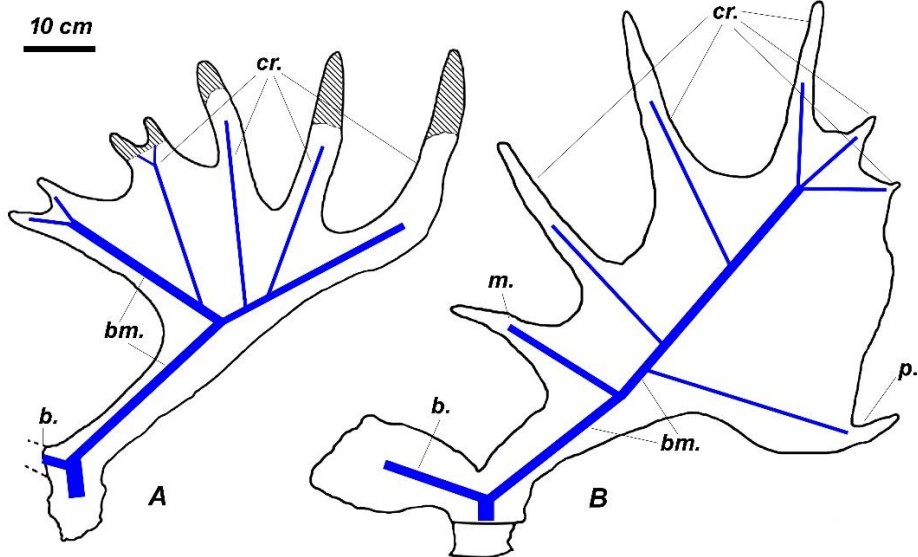

**Figure 12.** Comparison of antler bauplan in *Sinomegaceros stavropolensis* (**A**) and *Megaloceros giganteus* antecedens (**B**).

The supposed direct evolutionary relationship between *Praedama novocarthaginiensis* and *M. giganteus antecedens* [56] is improbable since the giant deer from Steinheim maintains the cursorial type of limbs, an old specialization that has been already lost by *P. novocarthaginiensis* (Figure 5B). The local evolution of *Praedama* in Europe is represented by the endemic small-sized species *P. matritensis* from the late Middle Pleistocene of the Iberian Peninsula and *P. savini* with simplified antlers without palmations from the Middle Pleistocene of Western Europe.

Titov and Shvyreva [105] established a new species *Megaloceros stavropoliensis* from the early Late Villafranchian of the Stavropol area (South of European Russia) that represents, according to the cited authors, a transitional evolutionary stage between *Rucervus* (*Arvernoceros*) *ardei* and *M. giganteus antecedens*. In my opinion, the antler from Stavropol shows a specific bauplan and shape that approaches it to Asian *Sinomegaceros*. Among the specific *Sinomegaceros* characteristics that should be mentioned are the well-expressed beam bending in the area of the first tine, the rather low insertion of the first tine, the relatively short beam segment with triangular cross-section between the first ramification and palmed portion and the fan-shaped palmation that forms a right angle with the anterior side of the above-mentioned beam portion [26,38,106]. The antler bauplan of *Sinomegaceros stavropoliensis* (Titov and Svyreva, 2016), and other species of this genus, is very peculiar and does not show any similarity with *Megaloceros* (Figure 12). According to the antler construction, *Megaloceros* and *Sinomegaceros* lineages diverged at the stage of two-tined antlers and, therefore, only the basal tine and the main beam of those two genera represent true homologies. The second ramification in *Sinomegaceros* evolved as an addition of the third strong tine on the posterior side of the beam at a short distance from the first ramification. The main beam in the area of the second ramification in *Sinomegaceros* turns sharply toward the anterior and often is interpreted as the first digit of the palmation [38]. The additional crown tines evolved between main axes of the distal bifurcation (Figure 12A) and thus explain the peculiar fan-like shape of palmation in *Sinomegaceros* that developed between the two main branches of distal bifurcation. Such a specific antler bauplan with a

low position of the second bifurcation had already evolved in *Praesinomegaceros venustus* from the Late Miocene of Central Asia [38,106]. *Arvernoceros insolitus* from Dmanisi [107] is another species from the Early Pleistocene of the Caucasus region that is characterized by typical for *Sinomegaceros* antler bauplan and therefore should be included in this genus as *Sinomegaceros insolitus* (Vekua et al., 2010). The similarity of the antler of *S. stavropoliensis* and *S. insolitus* with *Rucervus* (*A.*) *ardei* is superficial: in *R.* (*A.*) *ardei* and its Late Villafranchian descent *R.* (*A.*) *radulescui*, the distal palmation evolved as a fusion of crown tines (often bifurcated) situated on the posterior side of the beam at a long distance from the basal ramification [83].

*M. giganteus ruffii* is another woodland form that evolved from the dispersal of Asian giant deer in Western Europe by the beginning of the Late Pleistocene [25,26]. The ancestral form of *M. giganteus ruffii* had already evolved larger and more specialized antlers with a comparatively large span. Such initially advanced open habitat specialization required a more complicated transformation of antler shape that we can see in *M. giganteus ruffii*: its large distal crown portion became twisted and curved upright and somewhat medially, while the anterior crown tines were displaced distally and adjoin the series of distal crown tines, so the overall shape of the antler crown became compact and stream-lined. The metapodials became rather short that indicates a rapid loss of cursorial abilities. It is possible that the robust short-limbed type of giant deer from the Rhine Valley (Figure 5B) and the Early-Middle Devensian (approximately 115−25 kyr BP) robust giant deer from the British Isles [17] represent the further adaptation of local giant deer populations to forest habitats. The geographical distribution of *M. giganteus ruffii* was larger than in *M. giganteus antecedens* and possibly indicates the broader ecological tolerance of the former subspecies.

*M. giganteus padanus* is very interesting since it represents the local extreme special-ization of *M. giganteus ruffii* to woodland conditions in southern Europe. The antlers of *M. giganteus padanus* are very divergent and extremely shortened, representing obvious parallelism with *M. giganteus antecedens*. As in the latter subspecies, the Italian endemic giant deer maintains the well-developed long posterior tine, while its middle tine is very small and completely fused with the palmation. It is possible that the differentiation of the Italian subspecies from *M. giganteus ruffii* evolved quite early and is represented by Pohlig's [18] *Cervus* (*Euryceros*) *italiae*, which shows some features that may be considered as transitional between typical *M. giganteus ruffii* and *M. giganteus padanus*: the antlers of the *italiae* form are comparatively more divergent, while its middle tine often adjoins the palmation [18], as one can see in *M. giganteus padanus*. The final evolutionary specializa-tion of the Italian endemic giant deer, possibly, has been triggered by the isolation in the southern glacial refugium.

The Late Devensian giant deer from the British Isles represent the further stage of the general evolutionary trend of adaptation to the open landscape and seasonality. The upper molars become relatively larger and supplemented with a strong basal cingulum that could be a side effect of cranial pachyostosis [28]. *M. giganteus giganteus* is characterized by the combination of very long metapodials and the extremely large antler span that represent the most evolved adaptations to open habitats among *Megaloceros* and related lineages. Geist [7] regarded the giant deer as a specialized short-legged open-landscape cursorial form similar to steppe *Bison* or *Saiga*, but this is not the case. It seems that Geist [7] based his eco-morphological model only on the samples of short-legged woodland *Megaloceros* from Western Europe. As Lister [17] has already noted, the relative length of distal limb bones in giant deer is correlated with the vegetational or topographic environment: long-limbed populations inhabited open treeless habitats, while short-limbed forms evolved in wooded environments. *M. giganteus giganteus* maintained the specialization of the long-legged open landscape runner that was already evolved in *Praedama giulii*. This type of cursorial adaptation, rather, corresponds to the trotter-runner type according to the classification proposed by Geist [7]. Probably, *M. giganteus giganteus* corresponds to one of two subclades of clade 5 established by Rey-Iglesia et al. [30] based on genetic analysis of the Late Glacial samples from central and northwestern Europe (including Ireland) and the Late Glacial and

Holocene specimens from Eastern Europe and Western Siberia. This subclade represents the episodic post-Late Glacial Maximum colonization of northwestern Europe by a population of giant deer from the Russian glacial refugium [27,30].

*M. giganteus megaceros* is the last West European giant deer form that most probably corresponds to the second subclade of clade 5 that colonized northern and northwestern Europe after the Late Glacial Maximum from a south European glacial refugium [30]. Unlike *M. giganteus giganteus*, *M. giganteus megaceros* is characterized by reduced antler span due to the more or less upright position of the distal portion of the antler and shortened distal crown tines and shorter metacarpals. Some individuals (for instance, the skull from Barmeath Castle on Figure 6) attain the degree of antler crown compactness comparable to that of *M. giganteus ruffii* (Figure 4).

The overview of the giant deer subspecies proposed here, of course, does not cover the entire range of diversity of forms of giant deer. The question of the taxonomical status of eastern forms of giant deer remains open. Scheglova [22] reported the presence of both *M. giganteus giganteus* and *M. giganteus ruffii* in Eastern Europe, however, as she noted, not all specimens fit the diagnostic characters of those subspecies. Some of the findings mentioned by Scheglova [22], such as the giant deer forms from Pokrovskoe (Lower Volga, Russia) and Penza (Russia), show a very small antler span.

According to Alekseeva [62] and Vasiliev [108], the remains of giant deer from Western Siberia do not agree with the diagnostic characters of giant deer subspecies described from Western Europe (*M. giganteus giganteus* and *M. giganteus ruffii*). Thus, the remains of giant deer from the eastern part of its area of distribution require a special systematic study in future. Here, I propose an overview of the most important findings of giant deer from Eastern Europe and Asia.

The distal portion of the antlers described by Khomenko [109] from the bank outcrop of the Kalaus River (northern foothills of Greater Caucasus) is similar to the juvenile antlers of *M. giganteus* from Dornești (Romania) and belongs to a giant deer form similar to *M. giganteus giganteus*. The findings of giant deer forms with strongly divergent antlers as in the nominotypical subspecies seem to be common in Eastern Kazakhstan and Western Siberia. The specimen from Grigorievka (Kazakhstan) is characterized by strongly divergent antlers with long and straight distal crown tines as in *M. giganteus giganteus* [64]. However, unlike the type specimen from Dunleer, the antler segment between burr and middle tine in the deer from Kazakhstan is rather short. The upper tooth row in the specimen from Grigorievka maintains the primitive proportions with a relatively long premolar series (80.5%). Shpansky [65] ascribed to *M. giganteus giganteus* the almost complete skeleton КП-7191 of giant deer from Dzhambul (Eastern Kazakhstan). The antler shape of the giant deer from Dzhambul fits the diagnostic characters of the nominotypical subspecies from Europe well. However, as the specimen from Grigorievka, the Dzhambul deer is distinguished by the relatively long upper and lower premolar series (71.6% and 67.3% respectively), the relatively short distance between the antler burr burr and the middle tine, and the somewhat shorter metacarpals (Figure 5B). According to the $^{14}$C date reported by Van der Plicht et al. [14], the giant deer from Dzhambul is geologically rather old (43,600 yr BP) and this circumstance perfectly explains the presence of the archaic morphological features in the giant deer from Kazakhstan. The giant deer from Dzhambul is less specialized and certainly is not identical to *M. giganteus giganteus*, but possibly is closely related to the origin of *M. giganteus giganteus*. Shpansky [65] notes such a peculiar feature of the giant deer from Dzhambul as the relatively long posterior limb if compared to the anterior one. This feature may be interpreted as an adaptation to saltatorial locomotion, which is characteristic of forest and woodland ruminants and represents a different strategy of locomotion [7,65,110]. The antlered skull from Komissarovo (Kemerovo, Western Siberia, Russia) ascribed to *M. giganteus giganteus* [64] is characterized by the morphological features found in the specimens from Grigorievka and Dzhambul and most probably represent a similar form of giant deer.

Another specimen from Komissarovo, a skull with partially preserved antlers reported as *M. giganteus ruffii* [64], is characterized by antlers with little divergence. However, unlike the typical *ruffii* from Western Europe, the anterior crown tines in the Siberian form are in their initial position (not displaced distally) and the palmed portions of antlers are not twisted. The skull Nr. 582 from Grigorievka (northeastern Kazakhstan) ascribed to *M. giganteus ruffii* is characterized by more divergent antlers and rather long upper tooth row (approximately 160 mm) [64]. The relative length of the premolar series (PP/MM = 73.9%) is equivocal and falls within the range of variation of both samples from the Rhine Valley and Ireland. The metacarpals from Zhana-Aul (northeastern Kazakhstan) are quite short and gracile [67] as with the metacarpals of *M. giganteus ruffii* from Western Europe (Figure 5B). One can assume that the most of remains of *M. giganteus* from northeastern Kazakhstan and Western Siberia represent well-distinguished and as yet undescribed stages of giant deer evolution and local specializations and could be described as representing new subspecies.

### 4.3. Paleobiogeography of Giant Deer

The early evolutionary stages of the *Praedama–Megaloceros* lineage have evolved in Central Asia and the middle latitudes of the Western Siberian lowland. The evolutionary transition from forest to open woodland habitats most probably took place during the Late Pliocene [90,91]. It is possible that the most ancient cervid that belongs to the giant deer lineage is "*Elaphurus*" *eleonorae* from the Late Pliocene–Early Pleistocene of Kuruksai (Tajikistan), however, this assumption requires confirmation since the distal part of its antlers is poorly known. In particular, it is not clear if the deer from Kuruksai possessed the middle (trez) tine or its homology as *Metacervocerus* and *Dama* [37]. A cervid form similar to "*E.*" *eleonorae* could represent the evolutionary radiation that took place after the split between the *Dama* and *Praedama–Megaloceros* lineages. The further evolution of the *Dama* lineage took place in the Ponto-Mediterranean area of Western Eurasia [87].

Vislobokova [91] argued that the evolutionary transition from *Arvernoceros* to *Megaloceros* took place in the northern area of the Black Sea. According to the recently obtained data, the Early Pleistocene of southeastern Europe actually has yielded remains of *Rucervus* (*Arvernoceros*) that belong to a rather large (approximately 230 kg) and in some respects more advanced species *R.* (*A.*) *radulescui*, however, this endemic persistence of the Pliocene lineage in southeastern Europe could not overcome the end-Villafranchian faunal turnover and became extinct [85].

Unfortunately, the Asian remains of *Praedama* are still unknown. *Praedama giulii* is the earliest representative of the genus that dispersed into western Eurasia by the end of the Villafranchian. This cervid had already evolved all the features of a Palearctic open woodland dweller: the long cursorial limbs, the middle and posterior antler tines, and the weak mandibular pachyostosis. *P. giulii* did not meet biogeographic obstacles during the dispersals into the Iberian, Balkan and Italian Peninsulas. This implies that by the end of the Villafranchian the paleogeographic importance of dense forests in the Mediterranean area was minimal. The rapid dispersal of *P. giulii* into the Italian Peninsula is particularly interesting since it indicates a weakening of the mountain forest Dinaric paleozoogeographic filter that caused partial biogeographic isolation of the Peninsula during the early part of the Late Villafranchian [111]. *Praedama* successfully overcame the early-middle Pleistocene transition in Western Europe where it was represented by the forest-adapted *P. savini* from northwestern Europe and *P. matritensis* adapted to open dry savanna-like habitats of the Iberian Peninsula where it persisted until the late Middle Pleistocene [69,88]. It is possible that the easternmost known finding of *Praedama* is the basal fragment of antlers from the Middle Pleistocene of Adji-Eilas (Armenia) reported by Vereschagin [112] as *Dama* cf. *mesopotamica*.

The genus *Megloceros* represents the next stage of evolution marked by the adaptation to open woodland with strongly seasonal continental climate and irregular, but rather high primary ecological production [89]. *Megaloceros* dispersed at least three times into

Western Europe where it repeatedly evolved adaptations to forest conditions, giving rise to the subspecies *antecedens*, *ruffii*, and *megaceros*. The genus *Megaloceros* successfully dispersed into the Near East where it is presented by a rather small and short-limbed species *M. mugharensis* (di Stefano, 1996) from the Middle Pleistocene of Tabun [83,100]. The successful dispersal of the *Megaloceros* lineage into North Africa represents one of two exceptional instances (the second species is *Cervus elaphus*) when cervids entered the African continent. The African lineage of giant deer evolved into a very specialized rather small-sized (approximately 100 kg) species, *Megaceroides algericus,* characterized by strong cranial pachyostosis, rather weak small teeth, robust short limbs, and small palmated antlers that have lost their basal tine [99]. *M. algericus* was adapted to periaquatic habitats and, as *M. giganteus* from Ireland, fed on aquatic plants. *M. algericus* persisted until Holocene (6641 to 6009 cal yr BP) in the restricted area of northwestern Africa [113].

Successful dispersals of *Megaloceros giganteus* in eastern Asia are unknown. This eastward dispersal failure may be explained by the successful local evolutionary radiation of *Sinomegaceros* that evolved several open woodland and forest adapted large-sized species that may be regarded as ecological counterparts of the European forest giant deer and represent a good example of parallelism in antler shape with *M. giganteus antecedens* and *M. giganteus padanus*. The ecological opportunism and evolutionary flexibility was the main eco-evolutionary strategy of *Megaloceros* that enabled this cervid lineage to colonize destabilized ecosystems perturbed by the glaciation climate shifts. However, the ecological opportunism became a weak point of giant deer since it became a poor competitor when confronted with more specialized cervids.

Interestingly enough, the initially open-landscape dweller *M. giganteus* failed to disperse southward into the Iberian Peninsula during the Middle Pleistocene and evolved into the West European endemic form, *M. giganteus antecedens*, with secondary adaptations to forested habitats. The endemic Iberian form of *Praedama* that was already well-adapted to the local conditions was the most probable and unavoidable for the *M. giganteus* paleo-biogeographic obstacle. In Eastern Europe, the Late Pleistocene distribution of giant deer was generally limited by the Balkan and Greater Caucasus mountains [76,112] that acted as northern borders of glacial refugia of warmth-loving species, including cervids [112].

The origin of the Middle and Late Pleistocene Mediterranean insular dwarf cervids is often related with continental "giant deer" [114], however, none of the known endemic insular cervids shows any clear affinity with *M. giganteus*. Pohlig [115] described a small palmated antler from the assumed "Norfolk interglacial" layer of the Puntali Cave near Carini (Sicily) as a dwarf form of the giant deer *Cervus* (*Euryceros*) *messinae*. The length of the preserved part of the antler (the distalmost small part is missing) exceeds 23 cm. The diameter of the antler base is approximately 15 mm (measurements are calculated from the figures provided by Pohlig). The antler is characterized by a rather high position of small basal tine (approximately 3 cm from the burr), a strong second tine inserted on the anterior side of the beam that may be interpreted as a homology of middle (trez) tine, and a palmation-like extension between the main beam and the second tine. The antler surface, according to Pohlig [115], is pearled. The high position of basal tine and the pearled antler surface rule out the possible evolutionary relationship between *M. giganteus* and *C. messinae*. Azzaroli [114] included the deer from Carini Cave in *Megaceros* Owen, 1844 assuming a very broad understanding of this genus. In our opinion, *C. messinae* should be, rather, regarded for the moment as a species incertae sedis since its morphological features are still poorly understood [116]. It is possible that the origin of *C. messinae* is related to *Praeelaphus lyra* from the Early Villafranchian of the Lower Valdarno, Tuscany (work in progress). Two small-sized deer from the Middle and Late Pleistocene of southern Italy and Sicily that sometimes are regarded as dwarf giant deer are the endemic forms of fallow deer: *Dama calabriae* (Bonfiglio, 1978) and *Dama carburangelensis* (De Gregorio, 1925) respectively [82,117]. Another small-sized insular cervid *Praemegaceros* (*Nesoleipoceros*) *cazioti* from the Late Pleistocene of Corsica and Sardinia is phylogenetically related to the giant continental form *P.* (*N.*) *solilhacus* [116]. The complete antlers of *Candiacervus*

*devosi* van der Geer, 2018 [118] show a striking similarity with antlers of *Praemegaceros verticornis* or *P. obscurus* and maintain the essential antler elements of the continental forms, such as the dorsal tine, the middle tine, the posterior tine in the area of beam upright bending, and the distal portion of the beam that has lost its crown tines. Other species and forms of *Candiacervus* from Crete represent further stages of the insular specialization of *Praemegaceros*.

Lister and Stuart [15] argued that the distribution of giant deer in the Asian part of its range was greatly influenced by paleobiome geography and confined to a latitudinal zone of "boreal parkland" or "boreal steppe-woodland" situated in the southern part of the West Siberian Plane between steppe-tundra to the north and desert and semi-desert to the south. It is possible that this vast region also was the core area of *Megaloceros* evolution. Today, this zone is covered by temperate broadleaf forest in the north and temperate dry steppe in the south. In the easternmost part of giant deer distribution (Angara-Lena Plateau), the expansion of the taiga biome contributed to the extirpation of the giant deer in the Holocene [14,15].

The Last Glacial Maximum gap in the paleontological record of giant deer represents an unresolved paleobiogeographic problem since the Last Glacial refugia of this species are not identified yet [15]. It is possible that the long and robust metacarpal of giant deer from Duruitoarea Veche (Moldova) provides an indication of one such glacial refugium in southern Moldova, southern Romania, and northern Bulgaria.

The extinction of *M. giganteus* in northwestern Europe was triggered by the Younger Dryas cold pulse [9,11,13,15]. The longest persistence of *M. giganteus* (until mid-Holocene) is recorded in the middle latitudes of the European part of Russia that is plausibly considered as the "core" range of the species [15]. According to Gould [8], the extremely specialized large antlers in *M. giganteus* seriously reduced its potential success in thick forests. Lister and Stuart [15] argued that the extirpation of giant deer in its eastern area of distribution was caused by the loss of suitable "forest-steppe" habitats and the advancement of closed forests. However, the present study demonstrates that *M. giganteus* was very flexible from the evolutionary point of view and successfully adapted to the forested habitats in its western range of distribution several times. The Holocene specimens of giant deer from Sapozhok and Kamyshlov that represent the last giant deer population shortly before its extinction are already characterized by a rather compact crown with diminished antler span (Figure 4). Therefore, the hypothesis of giant deer extinction caused by "forest-steppe" habitat loss is not fully satisfactory. It is necessary to keep in mind that *M. giganteus* was not the only large cervid species of the middle latitudes of Eurasia that faced postglacial environmental changes. The second species, *Alces alces*, had roughly similar ecological requirements but, unlike *M. giganteus*, was better adapted to dense forests, deep snow, and bogs [112]. Therefore, one can assume that *M. giganteus* was simply outcompeted by *A. alces* in new environmental conditions. The remains of giant deer from the Mesolithic site of Yasnikolskoye (European part of Russia) is associated with *Alces alces* and *Equus ferus* [119–121]. The ecological competition with those two species in the new forested environment could have been fatal for *M. giganteus*.

## 5. Conclusions

The holotype of the giant deer *Megaloceros giganteus* identified for the first time (Blumenbach, 1799) allows us to give a precise definition for the nominotypical subspecies *M. giganteus giganteus* and to provide the systematic description of intraspecific diversity of giant deer from its western part of distribution. The nominotypical subspecies of giant deer is a specialized cursorial open-landscape dweller with long and robust metapodials and a very large antler span. *M. giganteus giganteus* is the most specialized open landscape form of the giant deer species and thus fully corresponds to the well-known image of this species. However, the overview of the giant deer subspecies, as yet unnamed forms, and closely related species reveal a quite complex picture of the evolutionary radiation that

characterizes *M. giganteus* as one of the most flexible from the Holarctic cervid evolutionary point of view.

The origin of *M. giganteus* is probably related to the genus *Metacervocerus* that dispersed from Southeast Asia to Europe during the Early Pleistocene. *Metacervocerus* was represented during the Pliocene and Early Pleistocene of Eurasia by several species with simple three-tined antlers similar to those in the modern *Axis axis*. Some representatives of the genus, such as *M. pardinensis* and *M. punjabiensis*, are characterized by specific features (cingulum in upper molars and relatively long braincase, respectively) that are found also in giant deer. The origin of the *Megaloceros* phylogenetic branch is related to the open wooded habitats of southwestern Siberia and northern Kazakhstan that represent the core area of giant deer evolution and geographic distribution. Multivariate analysis of diagnostic craniodental characters confirmed the distant relationship between *Dama* and *Megaloceros* and demonstrated the evolutionary relationship with *Praedama*. *Praedama* is a side evolutionary branch of *Megaloceros* lineage that dispersed in Europe by the end of the Villafranchian. Early representatives of *Praedama* already possess the antler bauplan typical for *Megaloceros* and show a weak development of mandibular pachyostosis. *Praedama giulii* (Kahlke, 1997) from the late Early Pleistocene of Germany is a specialized large cursorial deer with very long metapodials. This general evolutionary specialization was maintained in most Eurasian giant deer forms. Following the Pleistocene climate fluctuations, *M. giganteus* repeatedly dispersed into Western Europe and gave rise to the forest and woodland forms *M. giganteus antecedens*, *M. giganteus ruffii*, and *M. giganteus megaceros*. The advancing glaciations triggered the geographic isolations, southward dispersals and local endemic evolutionary processes that gave rise to such specialized forms as *M. giganteus padanus* (Vialli, 1939) from the Late Pleistocene of Italy, *M. mugharensis* (di Stefano, 1996) from the Middle Pleistocene of Near East, and *Megaceroides algericus* (Lydekker, 1890) from the Late Pleistocene–early Holocene of North Africa. The extreme evolutionary plasticity of giant deer antlers refutes the views that have dominated the long research history on the large antlers of giant deer as a harmful overspecialization that caused the extinction of *M. giganteus*. Ecological competition with *Alces alces* in the new postglacial conditions could have been one of the important factors in the extinction of *Megaloceros giganteus*.

**Funding:** This research was supported by the Sepkoski Grant (2017) of the Paleontological Society (USA).

**Institutional Review Board Statement:** Not applicable.

**Informed Consent Statement:** Not applicable.

**Acknowledgments:** I wish to extend my special thanks to Lord Bellew for kindly providing photographs of the giant deer specimens from Barmeath Castle that helped me to identify the original material used for the scientific description of the giant deer species. I would like to thank Reinhard Ziegler (the State Museum of Natural History of Stuttgart), Andrew Currant (the Natural History Museum of London), Ştefan Vasile (the University of Bucharest), and Christine Argot (the National Museum of Natural History, Paris) for the access provided to the paleontological collections under their care. I also thank three anonymous reviewers for their comments and suggestions that improved the quality of this paper.

**Conflicts of Interest:** The author declares no conflict of interest.

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
