# Peer review of "Taxonomy, Systematics and Evolution of Giant Deer Megaloceros Giganteus (Blumenbach, 1799) (Cervidae, Mammalia) from the Pleistocene of Eurasia"

_quaternary, doi:10.3390/quat4040036_

Round 1

Reviewer 1 Report

Reviewed article by R. Croitor “Taxonomy, systematics and evolution of giant deer Megaloceros giganteus (Blumenbach, 1799) (Cervidae, Mammalia) from the Pleistocene of Eurasia ” is the important scientific research. The author has collected previously published data and his own data on the morphology of the horns, the dentition, skulls and some elements of postcranial skeleton. The proportions of the metapodia were analyzed in relation to the overall body size in Megaloceros giganteus. The author used data on almost all significant and complete finds of giant deer remains in Western and Eastern Europe, as well as in the Caucasus. There were 29 of the morphological signs identified that are fundamental for constructing a cladogram of the systematic. The cladogram allows to visually reflect the phylogenetic relationships within the genus and establish the validity / non-validity of the subspecies identified earlier.

The author argues and substantiates the choice of the type specimen and type locality of Megaloceros giganteus (Blumenbach, 1799), which essential for determining the size of a group of giant deer. One of the few questions that arises when reading the manuscript is the author's determination of the differences between the Caucasian deer and the genus Megaloceros. The remains of deer from the Caucasus are fragmentary and only an analysis of the structure of the antlers is possible. Perhaps this is not enough to draw conclusions about their systematic affiliation. Nevertheless, the conclusion made by the author expresses his point of view on this problem.

Author Response

Thank you for your time and efforts while reviewing my manuscript. Concerning your question on Caucasian deer, I do agree that the material from this biogeographically complex region generally is incomplete or insufficiently described and many cervid remains require a more thorough study. Fortunately, the Caucasian region has yielded complete antlers that represent the essential part of cervid body that provides all necessary diagnostic characters and allows (with some rare exceptions of highly specialized species or genera with evolutionary conservative antlers, such as the telemetacarpal genus Procapreolus) to find the systematic position of a given cervid form. The cranial and dental morphology provides important diagnostic characters at the genus level (Heintz, 1970; Vislobokova, 1990). However, even if we have at our disposal complete craniodental data, it is often very difficult to assign our material to a species without knowing the antler morphology.

The large deer from Untermassfeld (Germany) discussed in the submitted paper is a very good illustration of this problem: the extremely rich postcranial and craniodental material from Untermassfeld shows that we are dealing with a large cervid that recently evolved in the Palearctic Realm and acquired some specialized features like large body size and long limbs, but still maintains the primitive morphology of dentition and skull. There are several genera of that epoch that share a similar degree of evolutionary specialization, including Eucladoceros and Rucervus (Arvernoceros). Kahlke (1997) described the deer from Untermassfeld as Eucladoceros according to the knowledge on cervid evolution at that time. My colleague and me (Croitor & Kostopoulos, 2004) included the deer from Untermassfeld in the genus Arvernoceros (now I regard it as a subgenus of Rucervus) because we were familiar with the material of Eucladoceros ctenoides from Ceyssaguet (almost coeval with Untermassfeld) that was clearly different from the deer from Undermassfeld and the juvenile antler from Untermassfeld does not correspond to the juvenile antlers of Eucladoceros. But I have to admit that we remained in the same frames of vision of the problem, lacking the information on the morphology of fully-grown antlers. Alas, the cranial and dental material was not of much help in this case. Finally, the new available material and old (and unfortunately forgotten) publications allow us to conclude, that we are dealing with a primitive representative of the Praedama-Megaloceros lineage that, as Eucladoceros and Rucervus (Arvernoceros), maintained the primitive and generalized craniodental morphology. Of course, it is possible to find good morphological diagnostic differences between Praedama, Eucladoceros, and Rucervus, but only after we perform the species and genus determinations based on the antler morphology. Me and my colleagues are often puzzled by a cervid material that contains cranial and dental material, but no antlers. Therefore, I believe that the systematic position of the Caucasian deer from Stavropol and Dmanisi based on the cranial morphology is robust. The antler bauplan (the general type of antler construction) of advanced cervids maintains all stages of evolution path made by the cervid lineage in question—from the simple three-pointed stage to the final degree of specialization—and permits to find the systematic position of a deer in question. The antler bauplan of large deer from Stavropol and Dmanisi is the same as in Asian Sinomegaceros as I have already demonstrated in the manuscript.

Reviewer 2 Report

The manuscript “Taxonomy, systematics and evolution of giant deer Megaloceros giganteus (Blumenbach, 1799) (Cervidae, Mammalia) from the Pleistocene of Eurasia” gives an interesting overview of the evolution of the genus Megaloceros and of different subspecies of Megaloceros giganteus. I would advise the author to include the genetic data that are available on this species in the Discussion:

Hughes, Sandrine, Thomas J. Hayden, Christophe J. Douady, Christelle Tougard, Mietje Germonpré, Anthony Stuart, Lyudmila Lbova, Ruth F. Carden, Catherine Hänni, Ludovic Say. 2006. Molecular Phylogeny of the Extinct Giant Deer, Megaloceros giganteus. Molecular Phylogenetics and Evolution 40, 285-291.

Immel, A., Drucker, D., Bonazzi, M. et al. Mitochondrial Genomes of Giant Deers Suggest their Late Survival in Central Europe. Sci Rep 5, 10853 (2015). https://doi.org/10.1038/srep10853

Rey-Iglesia Alba, Lister Adrian M., Campos Paula F., Brace Selina, Mattiangeli Valeria, Daly Kevin G., Teasdale Matthew D., Bradley Daniel G., Barnes Ian, Hansen Anders J. 2021. Exploring the phylogeography and population dynamics of the giant deer (Megaloceros giganteus) using Late Quaternary mitogenomes. Proc. R. Soc. B. 288: 20201864.

Author Response

Thank you very much for your time, attention and efforts while working with the manuscript of my paper. I also thank you for the suggested bibliographic sources on the genetic studies of Megaloceros giganteus that I find important and very useful. I have included all of them in the article.

Reviewer 3 Report

There are no description and discussion about the age variation of antlers of Megaloceros giganteus in ontogeny, especially whether there are significant differences in palmation? Does the age variation have a systematic and taxonomic significance? 

In the phylogenetic analysis, some genera and species closely related to Megaloceros are not discussed. Although Sinomegaceros is discussed and compared in the text, for example, it is not included in the phylogenetic analysis. In addition, the related taxa in the eastern distribution of Eurasia should be added. 

Alces alces, which has ecological competition with Megaloceros, has strong palmation of their antlers. There should be a more in-depth comparison between their palmation of the two forms.

Author Response

Thank you very much for your time and attention while working with the manuscript of my paper and for the thought-provoking questions. Here are the responses to your comments:

  1. “There are no description and discussion about the age variation of antlers of Megaloceros giganteus in ontogeny, especially whether there are significant differences in palmation? Does the age variation have a systematic and taxonomic significance?”

– the ontogenetic variation of giant deer from the British Isles was described by Reynolds (1925). We have discussed the problem of ontogenetic variation of giant deer antlers in our previous paper(Croitor, R., Stefaniak, K., PawĹ‚owska, K., Ridush, B., Wojtal, P. and Stach, M., 2014. Giant deer Megaloceros giganteus Blumenbach, 1799 (Cervidae, Mammalia) from Palaeolithic of Eastern Europe. Quaternary International, 326, pp.91-104) in the context of the determination of a juvenile antler from Romania compared to the type specimen of M. giganteus ruffii, which is also juvenile. Since then, I did not find new data on the ontogenetic development of antlers of giant deer, therefore I cannot add something new to the discussion that we have already published in 2014. Concerning the taxonomic importance of juvenile cervid specimens, I think that first of all this is a source of confusion and long synonymy lists when different ontogenetic stages of antler developments are described as different species or even genera. There are many examples and the most famous case is that of cervid remains from the Late Miocene of Taraclia described by Khomenko as three different genera and species (Cervavitus tarakliensis, Cervocerus novorossiae, and Damacerus bessarabiae), which represent different ontogenetic stages of antler development of the same species. Some authors tried to use the peculiarities of ontogenetic development of antlers as a diagnostic feature permitting to distinguish two genera with antlers that show a similar stage of evolutionary development. For example, Korotkevich (1970) argued that the three-pointed antlers of Procapreolus and Pliocervus are distinguished by the different order of antler tines development. This attempt to use the ontogenetic development of antlers as a diagnostic argument finally was finally rejected (Croitor & Stefaniak, 2009) since the samples of ontogenetic variants of modern deer (Capreolus, for instance) demonstrate that the antler tines may vary in the sequence of their appearance. Nonetheless, in some cases, ontogenetic stages of antler development, as well as rudiments and the individual variation may be useful for understanding how certain antler specializations evolved. But I do not suggest to use these sources of information for taxonomic purpose because of their instability. Of course, there are many things in antler development that we do not know yet. Possibly, there are some general patterns of development specific for certain phylogenetic lineages and mirroring the physiological peculiarities of evolutionary history of cervid groups (for instance, deer with cranial pachyostosis). But this study requires a large amount of data on the ontogenetic development of modern and fossil cervids. But, of course, this point was beyond the scope of the present study.

  1. “In the phylogenetic analysis, some genera and species closely related to Megaloceros are not discussed. Although Sinomegaceros is discussed and compared in the text, for example, it is not included in the phylogenetic analysis. In addition, the related taxa in the eastern distribution of Eurasia should be added”

Sinomegaceros is an interesting Asian genus that in my opinion represents a parallel or even a convergent evolution with Megaloceros. Both Sinomegaceros and Megaloceros evolved in the similar environmental conditions of the middle latitudes of Asia and evolved similar morphological and physiological adaptations (pachyostosis), but they are very distinct in antler bauplan that implies the large phylogenetic distance between Sinomegaceros and Megaloceros (see, for instance, our recent publication: Croitor, R., Abbas, S.G., Babar, M.A. and Khan, M.A., 2021. A new deer species (Cervidae, Mammalia) from the upper Siwaliks (Pakistan). Quaternary International, 595, pp.1-11). The evolutionary history of Sinomegaceros is traced from Praesinomegaceros of the Late Miocene of Central Asia (Vislobokova, 2009). As I demonstrated in the submitted paper, Megaloceros is a descent of Praedama and together with Dama is related to Metacervocerus with simple three-pointed antlers. Therefore, we are dealing with two different phylogenetic lineages: (1) PraesinomegacerosSinomegaceros and (2) MetacervocerusPraedamaMegaloceros. The large phylogenetic distance between Megaloceros and Sinomegaceros is confirmed by our cluster analysis of antler characters (Croitor, R., Abbas, S.G., Babar, M.A. and Khan, M.A., 2021. A new deer species (Cervidae, Mammalia) from the upper Siwaliks (Pakistan) …). Therefore, the addition of Sinomegaceros to the discussion, in my opinion, is out of the scope of this article: it does not bring any information on the systematics and taxonomy of Megaloceros and just will burden the discussion chapter of the article. Nonetheless, it could be very interesting to add the data on Sinomegaceros craniodental and antler characters to the cluster analysis. But, unfortunately, I could not do it, since the detailed cranial morphology of Sinomegaceros remains practically undescribed and I did not have the opportunity to work with the fossil material of Sinomegaceros.

  1. Alces alces, which has ecological competition with Megaloceros, has strong palmation of their antlers. There should be a more in-depth comparison between their palmation of the two forms”

– I think that Alces was a strong ecological competitor not only for Megaloceros in western Eurasia, but also for Sinomegaceros in Eastern Asia, and Cervalces in North America. As a result of this competition for the same ecological resources, Megaloceros, Sinomegaceros, and Cervalces went extinct. The evolution of antlers reflects first of all the social behaviour during the rutting period and some biological and ecological aspects of the rutting ( see Davitashvili, 1961 cited in the manuscript). Apparently, Megaloceros giganteus antecedens, M. giganteus padanus, Sinomegaceros, and A. alces arrived at a similar evolutionary compromise between the inherited features like the palmation, the rutting behaviour, and the environmental constraints. However, this interesting idea requires a special study analyzing and explaining the origin of the initial antler bauplan for Alces (a telemetacarpal deer), Sinomegaceros (a Late Miocene out-shoot of Cervinae), and Megaloceros that belongs to a group of Palearctic Cervidae with early acquisition of the middle (trez) tine. This is a complex problem that requires a special study with a different concept and a different methodological approach. Therefore, it is not possible to include this idea in the present article because without a good methodological and factual background it will remain speculative.